# Optimal Scalarizations for Sublinear Hypervolume Regret

**Qiuyi (Richard) Zhang**
Google Deepmind
`qiuyiz@google.com`

## Abstract

Scalarization is a general, parallizable technique that can be deployed in any multi-objective setting to reduce multiple objectives into one, yet some have dismissed this versatile approach because linear scalarizations cannot explore concave regions of the Pareto frontier. To that end, we aim to find simple non-linear scalarizations that provably explore a diverse set of $k$ objectives on the Pareto frontier, as measured by the dominated hypervolume. We show that hypervolume scalarizations with uniformly random weights achieves an optimal sublinear hypervolume regret bound of $O(T^{-1/k})$, with matching lower bounds that preclude any algorithm from doing better asymptotically. For the setting of multiobjective stochastic linear bandits, we utilize properties of hypervolume scalarizations to derive a novel non-Euclidean analysis to get regret bounds of $\widetilde{O}(dT^{-1/2} + T^{-1/k})$, removing unnecessary poly$(k)$ dependencies. We support our theory with strong empirical performance of using non-linear scalarizations that outperforms both their linear counterparts and other standard multiobjective algorithms in a variety of natural settings.

## 1  Introduction

Optimization objectives in modern AI systems are becoming more complex with many different components that must be combined to perform precise tradeoffs in machine learning models. Starting from standard $\ell_p$ regularization objectives in regression problems [Kutner et al., 2005] to increasingly multi-component losses used in reinforcement learning [Sutton et al., 1998] and deep learning [LeCun et al., 2015], many of these single-objective problems are phrased as a scalarized form of an inherently $k$-objective problem.

**Scalarization Method.** Practitioners often vary the weights of the scalarization method, with the main goal of exploring the entire *Pareto frontier*, which is the set of optimal objectives that cannot be simultaneously improved. First, one chooses some weights $\lambda \in \mathbb{R}^k$ and scalarization functions $s_\lambda(y) : \mathbb{R}^k \to \mathbb{R}$ that convert $k$ multiple objectives $F(a) := (f_1(a), ..., f_k(a))$ over some parameter space $a \in \mathcal{A} \subseteq \mathbb{R}^d$ into a single-objective scalar. Optimization is then applied to this family of single-objective functions $s_\lambda(F(x))$ for various $\lambda$ and since we often choose $s_\lambda$ to be monotonically increasing in all coordinates, $x_\lambda = \arg\max_{a \in \mathcal{A}} s_\lambda(F(a))$ is on the Pareto frontier and the various choices of $\lambda$ recovers an approximation to the Pareto frontier [Paria et al., 2018].

Due to its simplicity of use, many have turned to a heuristic-based scalarization strategy to pick the family of scalarizer and weights, which efficiently splits the multi-objective optimization into numerous single "scalarized" optimizations [Roijers et al., 2013]. Linear scalarizations with varying weights are often used in multi-objective optimization problems, such as in multi-objective reinforcement learning to combine task reward with the negative action norm [Abdolmaleki et al., 2021] or in RLHF to align responses with human preferences Ouyang et al. [2022]. However, the appeal of

38th Conference on Neural Information Processing Systems (NeurIPS 2024).

using scalarizations in multiobjective optimization declined as linear scalarizations are shown to be provably incapable of exploring the full Pareto frontier [Boyd and Vandenberghe, 2004].

**Beyond Linear Scalarization.** To address this, some works have proposed piecewise linear scalarizations inspired by economics [Busa-Fekete et al., 2017], while for multi-armed bandits, scalarized knowledge gradient methods empirically perform better with non-linear scalarizations [Yahyaa et al., 2014]. The classical Chebyshev scalarization has been shown to be effective in many settings, such as evolutionary strategies Qi et al. [2014], Li et al. [2016], general blackbox optimization [Kasimbeyli et al., 2019] or reinforcement learning Van Moffaert et al. [2013]. Other works have come up with novel scalarizations that perform better empirically in some settings [Aliano Filho et al., 2019, Schmidt et al., 2019]. There also have been specific multi-objective algorithms tailored to specific settings such as ParEgo [Knowles, 2006] and MOEAD [Zhang and Li, 2007] for black-box optimization or multivariate iteration for reinforcement learning [Yang et al., 2019]. Furthermore, many adaptive reweighting strategies have been proposed in order to target or explore the full Pareto frontier, which have connections to gradient-based multi-objective optimization [Lin et al., 2019, Abdolmaleki et al., 2021]. However these adaptive strategies are heuristic-driven and hard to compare, while understanding simple oblivious scalarizations remain very important especially in batched settings where optimizations are done heavily in parallel Gergel and Kozinov [2019].

**Hypervolume Regret.** To determine optimality, a natural and widely used metric to measure progress of an optimizer is the *hypervolume indicator*, which is the volume of the dominated portion of the Pareto set [Zitzler and Thiele, 1999]. The hypervolume metric has become a gold standard because it has strict Pareto compliance meaning that if set $A$ is a subset of $B$ and $B$ has at least one Pareto point not in $A$, then the hypervolume of $B$ is greater than that of $A$. Unsurprisingly, multiobjective optimizers often use hypervolume related metrics for progress tracking or acquisition function, such as the Expected Hypervolume Improvement (EHVI) or its differentiable counterpart [Daulton et al., 2020, Hupkens et al., 2015]. Only recently has some works provide sub-linear hypervolume regret bounds which guarantees convergence to the full Pareto frontier; however, they are exponential in $k$ and its analysis only applies to a specially tailored algorithm that requires an unrealistic classification step [Zuluaga et al., 2013]. We draw inspiration on work by [Golovin and Zhang, 2020] that introduces random hypervolume scalarizations but their work does not consider the finite-sample regime and their regret bounds are in the blackbox setting, where the convergence rate quantifies the statistical convergence of the Pareto frontier estimation.

## 1.1 Our Contributions

We show, perhaps surprisingly, that a simple ensemble of nonlinear scalarizations, known as Hypervolume scalarizations, are theoretically optimal to minimize hypervolume regret and are empirically competitive for general multiobjective optimization. Intuitively, we quantify how fast scalarizations can approximate the Pareto frontier under finite samples even with perfect knowledge of the Pareto frontier, which is the white-box setting. Specifically, as the hypervolume scalarization has sharp level curves, they naturally allow for the targeting of a specific part of the Pareto frontier, without any assumptions on the Pareto set or the need for adaptively changing weights. Additionally, we can obliviously explore the Pareto frontier by choosing $T$ maximizers of randomly weighted Hypervolume scalarizations and achieve a sublinear hypervolume regret rate of $O(T^{-1/k})$, where $T$ is the number of points sampled.

**Sublinear Hypervolume Regret.** Given any set of objectives $\mathbf{Y}$ that are explicitly provided, our goal is to choose points on the Pareto frontier in a way to maximize the hypervolume. In this white-box setting, we introduce the notion of the hypervolume regret convergence rate, which is both a function of both the scalarization and the weight distribution, and show that the maxima of Hypervolume scalarization with uniform i.i.d. weights enjoy $O(T^{-1/k})$ hypervolume regret (see Theorem 7). In fact, our derived regret rate of the Hypervolume scalarization holds for all frontiers, regardless of the inherent multiobjective function $F$ or the underlying optimizer. Therefore, we emphasize that analyzing these model-agnostic rates can be a general theoretical tool to compare and analyze the effectiveness of proposed multiobjective algorithms. Although many scalarizers will search the entire Pareto frontier as $T \to \infty$, the rate at which this convergence occurs can differ significantly, implying that this framework paves the road for a theoretical standard by which to judge the effectiveness of advanced strategies, such as adaptively weighted scalarizations.

**Lower bounds.** On the other hand, we show surprisingly that no multiobjective algorithm, whether oblivious or adaptive, can beat the optimal hypervolume regret rates of applying single-objective optimization with the hypervolume scalarization. To accomplish this, we prove novel lower bounds showing one cannot hope for a better convergence rate due to the exponential nature of our regret, for any set of $T$ points. Specifically, we show that the hypervolume regret of any algorithm after $T$ actions is at least $\Omega(T^{-1/k})$, demonstrating the necessity of the $O(T^{-1/k})$ term up to small constants in the denominator. As a corollary, we leverage the sublinear regret properties of hypervolume scalarization to transfer our lower bounds to the more general setting of scalarized Bayes regret. Together, we demonstrate that for general multiobjective optimzation, finding maximas of the hypervolume scalarizations with a uniform weight distribution optimally finds the Pareto frontier asymptotically.

**Theorem 1** (Informal Restatement of Theorem 7 and Theorem 8). *Let $\mathbf{Y}_T = \{y_1, ..., y_T\}$ be a set of $T$ points in $\mathbb{R}^k$ such that $y_i \in \arg\max\limits_{y \in \mathcal{Y}} s_{\lambda_i}^{\mathrm{HV}}(y)$ with $\lambda_i \sim \mathcal{S}_+$ randomly drawn i.i.d. from an uniform distribution and $s^{\mathrm{HV}}$ are hypervolume scalarizations. Then, the hypervolume regret satisfies*

$$\mathcal{HV}(\mathcal{Y}^\star) - \mathcal{HV}(\mathbf{Y}_T) = O(T^{-\frac{1}{k}})$$

*where $\mathcal{Y}^\star$ is the Pareto frontier and $\mathcal{HV}$ is the hypervolume function. Furthermore, any algorithm for choosing these $T$ points must suffer hypervolume regret of at least $\Omega(T^{-\frac{1}{k}})$.*

**Scalarized Algorithm for Linear Bandit.** Next, we use a novel non-Euclidean analysis to prove improved hypervolume regret bounds for our theoretical toy model: the classic *stochastic linear bandit* setting. For any scalarization and weight distribution, we propose a new scalarized algorithm (Algorithm 1) for multiobjective stochastic linear bandit that combines uniform exploration and exploitation via an UCB approach to provably obtain scalarized Bayes regret bounds, which we then combine with the hypervolume scalarization to derive optimal hypervolume regret bounds. Specifically, for any scalarization $s_\lambda$, we show that our algorithm in the linear bandit setting has a scalarized Bayes regret bound of $\widetilde{O}(L_p k^{1/p} d T^{-1/2} + T^{-1/k})$, where $L_p$ is the Lipschitz constant of the $s_\lambda(\cdot)$ in the $\ell_p$ norm. Finally, by using hypervolume scalarizations and exploiting their $\ell_\infty$-smoothness, we completely remove the dependence on the number of objectives, $k$, which had a polynomial dependence in previous regret bound given by Golovin and Zhang [2020].

**Theorem 2** (Informal Restatement of Theorem 11). *For the multiobjective linear bandit problem, let $\mathbf{A}_T \subseteq \mathcal{A}$ be the actions generated by $T$ rounds of Algorithm 1, then our hypervolume regret is bounded by:*
$$\mathcal{HV}_z(\Theta^\star \mathcal{A}) - \mathcal{HV}_z(\Theta^\star \mathbf{A}_T) \leq O(d T^{-\frac{1}{2}} + T^{-\frac{1}{k}})$$

**Experiments.** Guided by our theoretical analysis, we empirically evaluate a diverse combination of scalarizations and weight distributions in multiple natural settings. First, we consider a synthetic optimization that is inspired by our theoretical derivation of hypervolume regret when the entire Pareto front is known and analytically given. Thus we can explicitly calculate hypervolume regret as a function of the number of Pareto points chosen and compare the regret convergence rates of multiple scalarizations with the uniform $\mathcal{S}_+$ distribution with $k = 3$ objectives, for easy visualization. This is important since when $k = 2$, we show that the Chebyshev and Hypervolume scalarizations are in fact equivalent.

In our setup, we consider synthetic combinations of concave, convex, and concave/convex Pareto frontiers in each dimension. As expected, non-linear Hypervolume and Chebyshev scalarizations enjoy fast sublinear convergence, while the performance of the Linear scalarization consistently lacks behind, surprisingly even for convex Pareto frontiers. We observe that generally the Hypervolume scalarization does better on concave Pareto frontiers and on some convex-concave frontiers, while the Chebyshev distribution have faster hypervolume convergence in certain convex regimes.

For multiobjective linear bandits, our experiments show that for many settings, despite having a convex Pareto frontier, applying linear or Chebyshev scalarizations naively with various weight distributions leads to suboptimal hypervolume progress, especially when the number of objective increase to exceed $k \geq 5$. This is because the non-uniform curvature of the Pareto frontier, exaggerated by the curse of dimensionality and combined with a stationary weight distribution, hinders uniform progress in exploring the frontier. Although one can possibly adapt the weight distribution to the varying curvature of the Pareto frontier when it is convex, the use of simple non-linear scalarizations allow for fast parallelization and are theoretically sound.

For general multiobjective optimization, we perform empirical comparisons on BBOB benchmarks for bi-ojective functions in a bayesian optimization setting, using classical Gaussian Process models [Williams and Rasmussen, 2006]. When comparing EHVI with hypervolume scalarization approaches, we find that EHVI tends to limit its hypervolume gain by over-focusing on the central portion of the Pareto frontier, whereas the hypervolume scalarization encourages a diverse exploration of the extreme ends.

We summarize our contributions as follows:

- Show that hypervolume scalarizations optimizes for the full Pareto frontier and enjoys sublinear hypervolume regret bounds of $O(T^{-1/k})$, a theoretical measure of characterizing scalarization effectiveness in the white-box setting.
- Establish a tight lower bound on the hypervolume regret for any algorithm and Bayes regret of $\Omega(T^{-1/k})$ by a packing argument on the Pareto set.
- Introduce optimization algorithm for multiobjective linear bandits that achieves improved $\widetilde{O}(dT^{-1/2} + T^{-1/k})$ hypervolume regret via a novel non-Euclidean regret analysis and metric entropy bounds.
- Empirically justify the adoption of hypervolume scalarizations for finding a diverse Pareto frontier in general multiobjective optimization via synthetic, linear bandit, and blackbox optimization benchmarks.

## 2  Problem Setting and Notation

For a scalarization function $s_\lambda(x)$, $s_\lambda$ is $L_p$-Lipschitz with respect to $x$ in the $\ell_p$ norm on $\mathcal{X} \subseteq \mathbb{R}^d$ if for $x_1, x_2 \in \mathcal{X}$, $|s_\lambda(x_1) - s_\lambda(x_2)| \leq L_p\|x_1 - x_2\|_p$, and $L_\lambda$ analogously for $\lambda$ in the Euclidean norm. We let $\mathcal{S}_+^{k-1} = \{y \in \mathbb{R}^k \,|\, \|y\| = 1, y > 0\}$ be the sphere in the positive orthant and by abuse of notation, we also let $y \sim \mathcal{S}_+^{k-1}$ denote that $y$ is drawn uniformly on $\mathcal{S}_+^{k-1}$. Our usual settings of the weight distribution $\mathcal{D} = \mathcal{S}_+^{k-1}$ will be uniform, unless otherwise stated.

For two outputs $y, z \in \mathcal{Y} \subseteq \mathbb{R}^k$, we say that $y$ is *Pareto-dominated* by $z$ if $y \leq z$ and there exists $j$ such that $y_j < z_j$, where $y \leq z$ is defined for vectors element-wise. A point is *Pareto-optimal* if no point in the output space $\mathcal{Y}$ can dominates it. Let $\mathcal{Y}^\star$ denote the set of Pareto-optimal points (objectives) in $\mathcal{Y}$, which is also known as the *Pareto frontier*. Our main progress metrics for multiobjective optimization is given by the standard hypervolume indicator. For $S \subseteq \mathbb{R}^k$ compact, let $\text{vol}(S)$ be the regular hypervolume of $S$ with respect to the standard Lebesgue measure.

**Definition 3.** *For a set $\mathcal{Y}$ of points in $\mathbb{R}^k$, we define the (dominated)* **hypervolume indicator** *of $\mathcal{Y}$ with respect to reference point $z$ as:*

$$\mathcal{HV}_z(\mathcal{Y}) = \text{vol}(\{x \,|\, x \geq z, x \text{ is dominated by some } y \in \mathcal{Y}\})$$

We can formally phrase our optimization objective as trying to rapidly minimize the hypervolume (psuedo-)regret. Let $\mathcal{A}$ be our action space and for some general multi-objective function $F$, let $\mathcal{Y}$ be the image of $\mathcal{A}$ under $F$. Let $\mathbf{A}_T \in \mathbb{R}^{T \times d}$ be any matrix of $T$ actions and let $\mathbf{Y}_T = F(\mathbf{A}_T) \in \mathbb{R}^{T \times k}$ be the $k$ objectives corresponding. Then, the hypervolume regret of actions $\mathbf{A}_T$, with respect to the reference point $z$, is given by:

$$\text{Hypervolume-Regret}(\mathbf{A}_T) = \mathcal{HV}_z(\mathcal{Y}^\star) - \mathcal{HV}_z(\mathbf{Y}_T)$$

which is $0$ for all $z$ if and only if $\mathbf{Y}_T$ contains all unique points in $\mathcal{Y}^\star$.

For various scalarizations and weight distributions, an related measure of progress that attempts to capture the requirement of diversity in the Pareto front is scalarized Bayes regret for some scalarization function $s_\lambda$. For some fixed scalarization with weights $\lambda$, $s_\lambda : \mathbb{R}^k \to \mathbb{R}$, we can define the instantaneous scalarized (psuedo-)regret as $r(s_\lambda, a_t) = \max_{a \in \mathcal{A}} s_\lambda(F(a)) - s_\lambda(F(a_t))$. To capture diversity and progress, we will vary $\lambda \sim \mathcal{D}$ according to some distribution of non-negative weight vectors and define regret with respect with respect to all past actions $\mathbf{A}_t$. Specifically, we define the *(scalarized) Bayes regret* with respect to a set of actions $\mathbf{A}_t$ to be: $BR(s_\lambda, \mathbf{A}_t) = \mathbf{E}_{\lambda \sim \mathcal{D}}[\max_{a \in \mathcal{A}} s_\lambda(F(a)) - \max_{a \in \mathbf{A}_t} s_\lambda(F(a))] = \mathbf{E}_{\lambda \sim \mathcal{D}}[\min_{a \in \mathbf{A}_t} r(s_\lambda, a)]$

Unlike previous notions of Bayes regret in literature, we are actually calculating the Bayes regret of a reward function that is maximized with respect to an entire set of actions $\mathbf{A}_t$. Specifically, by maximizing over all previous actions, this captures the notion that during multi-objective optimization our Pareto set is always expanding. We will see later that this novel definition is the right one, as it generalizes to the multi-objective setting in the form of hypervolume regret.

### 2.1 Scalarizations for Multiobjective Optimization

For multiobjective optimization, we generally only consider *monotone* scalarizers that have the property that if $y > z$, then $s_\lambda(y) > s_\lambda(z)$ for all $\lambda$. Note this guarantees that an optimal solution to the scalarized optimization is on the Pareto frontier. A common scalarization used widely in practice is the linear scalarization: $s_\lambda^{\text{LIN}}(y) = \lambda^\top y$ for some chosen positive weights $\lambda \in \mathbb{R}^k$. By Lagrange duality and hyperplane separation of convex sets, one can show that any convex Pareto frontier can be characterized fully by an optimal solution for some weight settings.

However, linear scalarizations cannot recover the non-convex regions of Pareto fronts since the linear level curves can only be tangent to the Pareto front in the protruding convex regions (see Figure 1). To overcome this drawback, another scalarization that is proposed is the Chebyshev scalarization: $s_\lambda^{\text{CS}}(y) = \min_i \lambda_i y_i$. Indeed, one can show that the sharpness of the scalarization, due to its minimum operator, can discover non-convex Pareto frontiers.

**Proposition 4** (Emmerich and Deutz [2018]). *For a point $y^\star \in \mathcal{Y}^\star$ for convex $\mathcal{Y}$, there exists $\lambda > 0$ such that $y^\star = \arg\max_{y \in \mathcal{Y}} s_\lambda^{\text{LIN}}(y)$. For a point $y^\star \in \mathcal{Y}^\star$ for any possibly non-convex set $\mathcal{Y}$ that lies in the positive orthant, there exists $\lambda > 0$ such that $y^\star = \arg\max_{y \in \mathcal{Y}} s_\lambda^{\text{CS}}(y)$.*

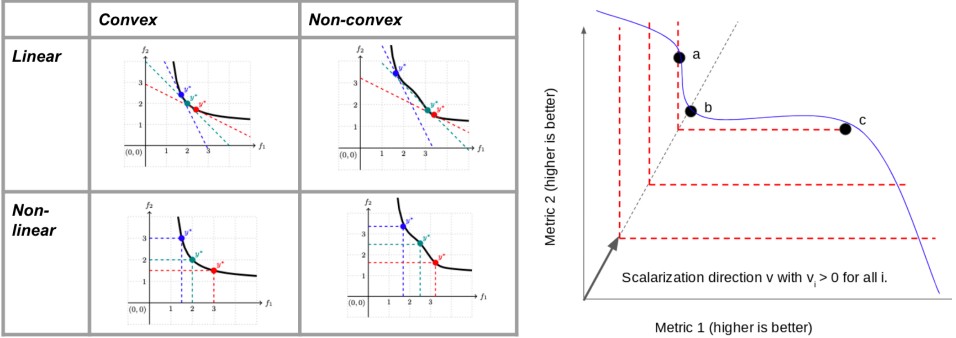

Figure 1: **Left:** Comparisons of the scalarized minimization solutions with various weights with convex and non-convex Pareto fronts. The colors represent different weights; the dots are scalarized optima and the dotted lines represent level curves. Linear scalarization does not have an optima in the concave region of the Pareto front for any set of weights, but the non-linear scalarization, with its sharper level curves, can discover the whole Pareto front (Figure from [Emmerich and Deutz, 2018]). **Right:** The dotted red lines represent the level curves of the hypervolume scalarization with $\lambda = v$, discovering $b$, whereas the linear scalarization would prefer $a$ or $c$. Furthermore, the optima is exactly the Pareto point that is in the direction of $v$.

## 3 Hypervolume Scalarizations

In this section, we show the utility and optimality of a related scalarization known as the Hypervolume scalarization, $s_\lambda^{\text{HV}}(y) = \min_i (y_i/\lambda_i)^k$ that was introduced in Golovin and Zhang [2020].

To gain intuition, we visualize the non-linear level curves of the scalarization, which shows that our scalarization targets the portion of the Pareto frontier in the direction of $\lambda$ for any $\lambda > 0$ (see Figure 1), since the tangent point of the level curves of the scalarization is always on the vector in the direction of $\lambda$. This implies that with an uniform distribution on $\lambda$, we are guaranteed to have a uniform spread of Pareto points in terms of its angular direction.

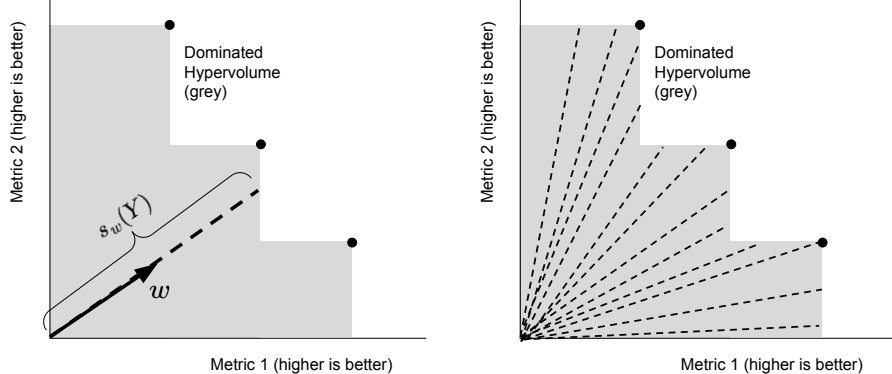

Figure 2: The hypervolume scalarization taken with respect to a direction $\lambda = w$ corresponds to a differential area element within dominated hypervolume and averaging over random directions is analogous to integrating over the dominated hypervolume in polar coordinates. We exploit this fact to show that by choosing the maximizers of $T$ random hypervolume scalarizers, we quickly converge to the hypervolume of the Pareto frontier at an optimal rate of $O(T^{-1/k})$. Figure from [Song et al., 2024]

**Lemma 5.** *For any point $y^\star$ on the Pareto frontier of any set $\mathcal{Y}$ that lies in the positive orthant, there exists $\lambda > 0$ such that $y^\star = \arg\max_{y \in \mathcal{Y}} s_\lambda^{\mathrm{HV}}(y)$. Furthermore, for any $\alpha, \lambda > 0$ such that $\alpha\lambda$ is on the Pareto frontier, then $\alpha\lambda \in \arg\max_{y \in \mathcal{Y}} s_\lambda^{\mathrm{HV}}(y)$.*

This scalarization additionally has the special property that the expected maximized scalarized value under a uniform weight distribution on $\mathcal{S}_+^{k-1}$ gives the dominated hypervolume, up to a constant scaling factor. Thus, the optima of the hypervolume scalarization over some static uniform distribution will be provably sufficiently diverse for any Pareto set in expectation.

**Lemma 6** (Hypervolume in Expectation [Golovin and Zhang, 2020])**.** *Let $\mathbf{Y}_T = \{y_1, ..., y_T\}$ be a set of $T$ points in $\mathbb{R}^k$. Then, the hypervolume with respect to a reference point $z$ is given by:*

$$\mathcal{HV}_z(\mathbf{Y}_T) = c_k \mathop{\mathbf{E}}_{\lambda \sim \mathcal{S}_+^{k-1}} \left[ \max_{y \in \mathbf{Y}_T} s_\lambda^{\mathrm{HV}}(y - z) \right]$$

*where $c_k = \frac{\pi^{k/2}}{2^k \Gamma(k/2+1)}$ is a constant depending only on $k$.*

While this lemma is useful in the infinite limit, we supplement it by showing that finite-sample bounds on the strongly sublinear hypervolume convergence rate. In fact, many scalarizations will eventually explore the whole Pareto frontier in the infinite limit, but the rate at which the exploration improves the hypervolume is not known, and may be exponentially slow. We show that the optimizing hypervolume scalarizations with a uniform weight distribution enjoys *sublinear hypervolume regret*, specifically $O(T^{-1/k})$ hypervolume regret convergence rates for any Pareto set $\mathcal{Y}$. Note that this rate is agnostic of the underlying optimization algorithm or Pareto set, meaning this is a general property of the scalarization.

Our novel proof of convergence uses a generalization argument to connect hypervolume-scalarized Bayes regret and its finite sample form, exploiting the Lipschitz properties of $s_\lambda^{\mathrm{HV}}$ to derive metric entropy bounds. Proving smoothness properties of our hypervolume scalarizations for any $\lambda > 0$ with $\lambda$ normalized on the unit sphere is non-obvious as $s_\lambda^{\mathrm{HV}}(y)$ depends inversely on $\lambda_i$ so when $\lambda_i$ is small, $s_\lambda^{\mathrm{HV}}$ might change wildly.

**Theorem 7** (Sublinear Hypervolume Regret)**.** *Let $\mathbf{Y}_T = \{y_1, ..., y_T\}$ be a set of $T$ points in $\mathbb{R}^k$ such that $y_i \in \arg\max_{y \in \mathcal{Y}} s_{\lambda_i}^{\mathrm{HV}}(y - z)$ with respect to a reference point $z$ and $B_l \le y_i - z \le B_u$. Then, with probability at least $1 - \delta$ over $\lambda_i \sim \mathcal{S}_+$ i.i.d., the hypervolume of $\mathbf{Y}_T$ with respect to $z$ satisfies*

*sublinear hypervolume regret in $T$:*

$$\mathcal{HV}_z(\mathcal{Y}^\star) - \mathcal{HV}_z(\mathbf{Y}_T) = O(T^{-\frac{1}{k+1}} + \sqrt{\ln(1/\delta)}T^{-\frac{1}{2}})$$

*where $O$ hides constant factors in $k, B_u, B_l$.*

*For $k = 2$, this also holds when Chebyshev scalarization is used: $y_i \in \arg\max\limits_{y \in \mathcal{Y}} s_{\lambda_i}^{\mathrm{CS}}(y - z)$.*

We note that the distribution of Pareto points selected by the Chebyshev scalarizer is quite similar to the Hypervolume scalarizer as the formulas are almost identical, except for the inverse weights of the latter. In fact, we show that when $k = 2$, both scalarizers behave the same and enjoy strong convergence rates; however for $k > 2$, the Pareto distributions are in fact different under $\lambda \sim \mathcal{S}_+$ and we can empirically observe the suboptimality of the Chebyshev scalarizer. By definition, using the inverse weight distribution with the Chebyshev scalarizer will be equivalent to applying the Hypervolume scalarizer.

## 3.1  Lower Bounds and Optimality

The dominating factor in our derived convergence rate is the $O(T^{-1/(k+1)})$ term and we show that this cannot be improved. Over all subsets $\mathbf{Y}_T \subseteq \mathcal{Y}$ of size $T$, note that our optimal convergence rate is given by the the subset that maximizes the dominated hypervolume of $\mathbf{Y}_T$, although finding this is in fact a NP-hard problem due to reduction to set cover. By constructing a lower bound via a novel packing argument, we show that even this optimal set would incur at least $\Omega(T^{-1/k})$ regret, implying that our convergence rates, derived from generalization rates when empirically approximating the hypervolume, are optimal.

Specifically, we show that for hypervolume regret, any algorithm cannot achieve better than $O(T^{-1/(k-1)})$ regret even when using linear objectives, and this matches the dominating factor in our algorithm up to a small constant in the denominator. By using hypervolume scalarizations and its connection to hypervolume regret, we conclude that this also implies a $\Omega(T^{-1/(k-1)})$ lower bound on the scalarized Bayes regret.

**Theorem 8** (Hypervolume Regret Lower Bound). *There exists a setting of linear objective parameters $\Theta^\star$ and $\mathcal{A} = \{a : \|a\| = 1\}$ such that for any actions $\mathbf{A}_T$, the hypervolume regret at $z = 0$ after $T$ rounds is*

$$\mathcal{HV}_z(\Theta^\star\mathcal{A}) - \mathcal{HV}_z(\Theta^\star\mathbf{A}_T) = \Omega(T^{-\frac{1}{k-1}})$$

**Corollary 9.** *There is a setting of objectives $\Theta^\star$ and $\mathcal{A} = \{a : \|a\| = 1\}$ such that for any actions $\mathbf{A}_T$, the scalarized Bayes regret after $T$ rounds is*

$$BR(s_\lambda^{\mathrm{HV}}, \mathbf{A}_T) = \Omega(T^{-\frac{1}{k-1}})$$

## 4  Multiobjective Stochastic Linear Bandits

We propose a simple scalarized algorithm for linear bandits and provide a novel $\ell_p$ analysis of the hypervolume regret that removes the polynomial dependence on $k$ in the scalarized regret bounds. When combined with the $\ell_\infty$ sharpness of the hypervolume scalarization, this analysis gives an optimal $O(d/\sqrt{T})$ bound on the scalarized regret, up to $\log(k)$ factors. This $\log$ dependence on $k$ is perhaps surprising but is justified information theoretically since each objective is observed separately. Note that our scalarized algorithm works despite of noise in the observations, which makes it difficult to even statistically infer measures of hypervolume progress. Our setup and algorithm is given in the Appendix (see Section A).

By using the confidence ellipsoids given by the UCB algorithm, we can determine each objective parameter $\Theta_i^\star$, up to a small error. To bound the scalarized regret, we utilize the $\ell_p$ smoothness of $s_\lambda$, $L_p$, to reduce the dependence on $k$ to be $O(k^{1/p})$, which effectively removes the polynomial dependence on $k$ when $p \to \infty$. This is perhaps not surprising, since each objective is observed independently and fully, so the information gain scales with the number of objectives.

**Lemma 10.** *Consider running* EXPLOREUCB *(Algorithm 1) for* $T > \max(k, d)$ *iterations and for* $T$ *even, let* $a_T$ *be the action that maximizes the scalarized UCB in iteration* $T/2$. *Then, with probability at least* $1 - \delta$, *the instantaneous scalarized regret can be bounded by*

$$r(s_\lambda, a_T) \leq 10 k^{\frac{1}{p}} L_p d \sqrt{\frac{\log(k/\delta) + \log(T)}{T}}$$

*where* $L_p$ *is the* $\ell_p$*-Lipschitz constant for* $s_\lambda(\cdot)$.

Finally, we connect the expected Bayes regret with the empirical average of scalarized regret via uniform convergence properties of all functions of the form $f(\lambda) = \max_{a \in \mathbf{A}} s_\lambda(\Theta^\star a)$. By using $s_\lambda^{\text{HV}}$ and setting $p = \infty$, we derive our final fast hypervolume regret rates for stochastic linear bandits, which is the combination of the scalarized regret rates and the hypervolume regret rates.

**Theorem 11** (HyperVolume Regret of EXPLOREUCB). *Let* $z \in \mathbb{R}^k$ *be a reference point such that over all* $a \in \mathcal{A}$, $B_l \leq \Theta^\star a - z \leq B_u$. *Then, with constant probability, running Algorithm 1 with* $s_\lambda^{\text{HV}}(y)$ *and* $\mathcal{D} = \mathcal{S}_+$ *gives hypervolume regret bound, for* $k$ *constant,*

$$\mathcal{HV}_z(\Theta^\star \mathcal{A}) - \mathcal{HV}_z(\Theta^\star \mathbf{A}_T) \leq O\left(d\sqrt{\frac{\log(T)}{T}} + \frac{1}{T^{\frac{1}{k+1}}}\right)$$

## 5 Experiments

In this section, we empirically justify our theoretical results by comparing hypervolume convergence curves for multiobjective optimization in synthetic, linear bandit and blackbox optimization environments with multiple scalarizations and weight distributions. Our empirical results highlight the advantage of the hypervolume scalarization with uniform weights in maximizing the diversity and hypervolume of the resulting Pareto front when compared with other scalarizations and weight distributions, especially when there are a mild number of output objectives $k$. Our experiments are not meant to show that scalarizations is the best way to solve multiobjective optimization; rather, it is a simple yet competitive baseline that is easily parallelized in a variety of settings. Also, we use slightly altered form of our hypervolume scalarization as $s_\lambda(y) = \min_i y_i / \lambda_i$, which is a simply a monotone transform and does not inherently affect the optimization. All error bars are given between the 30 to 70 percentile over independent repeats.

### 5.1 Synthetic Optimization

Our synthetic optimization benchmarks assume the knowledge of $\mathcal{Y}$ and the Pareto frontier and thus we can compute the total hypervolume and compare the hypervolume regret of multiple scalarizations with the uniform $\mathcal{S}_+$ distribution. For our experiments, we fix our weight distribution and compare the three widely types of scalarizations that were previously mentioned: the Linear, Chebyshev, and the Hypervolume scalarization. We focus on the $k = 3$ setting and apply optimization for a diverse set of Pareto frontiers in the region $x, y \in [0, 1], z > 0$. We discretize our region into 30 points per dimension to form a discrete Pareto frontier and set our reference point to be at $z = [-\epsilon, -\epsilon, -\epsilon]$ for $\epsilon = 1\text{e-}4$ and run for 10 repeats.

The synthetic Pareto frontiers that we are consider are a product of 1-dimensional frontiers and specifically, $z = g(x) \cdot g(y)$, where $g(x) = \exp(-x), 3 - \exp(x), \cos(\pi x) + 1$, which form a concave, convex, and concave/convex Pareto frontiers respectively. Now that since the derivatives of these functions are all negative in our feasible region, $z$ is a valid Pareto front. From our combination of functions, we observe that both Hypervolume and Chebyshev scalarizations enjoy fast convergence, while the performance of the Linear scalarization consistently lacks behind, surprisingly even for convex Pareto frontiers (see full plots in C.1). Interestingly, we observe that generally the Hypervolume scalarization does better on concave Pareto frontiers and on some convex-concave frontiers; however since these are static algorithms, it is not surprising that the Chebyshev distribution can perform better in certain convex regimes. However, we still advocate the usage of the hypervolume indicator as it is guaranteed to have a uniform spread especially when the number of objectives is increased, as shown by our later experiments.

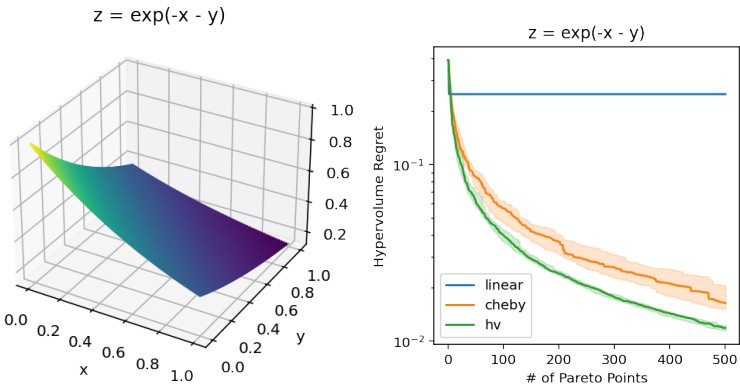

Figure 3: Comparisons of multiple scalarizations for the synthetic concave Pareto frontier given by $z = \exp(-x - y)$. The hypervolume regret for Linear is constant, and the Hypervolume enjoys a faster regret convergence rate than the Chebyshev.

## 5.2 Stochastic Linear Bandits

We run Algorithm 1 and compare scalarization effects for the multiobjective linear bandit setting. We set our reference point to be $\mathbf{z} = -\mathbf{2}$ in $k$ dimension space, since our action set of $\mathcal{A} = \{a : \|a\| = 1\}$ and our norm bound on $\Theta^\star$ ensures that our rewards are in $[-1, 1]$.

In conjunction with the scalarizer, we use our weight distribution $\mathcal{D} = \mathcal{S}_+$, which samples vectors uniformly across the unit sphere. In addition, we also compare this with the bounding box distribution methods that were suggested by [Paria et al., 2018], which samples from the uniform distribution from the min to the max each objective and requires some prior knowledge of the range of each objective [Hakanen and Knowles, 2017]. Given our reward bounds, we use the bounding box of $[-1, 1]$ for each of the $k$ objectives. Following their prescription for weight sampling, we draw our weights for the linear and hypervolume scalarization uniformly in $[1, 3]$ and take an inverse for the Chebyshev scalarization. We name this the boxed distribution for each scalarization, respectively.

To highlight the differences between the multiple scalarizations, we configure our linear bandits parameters to be anti-correlated, which creates a convex Pareto front with non-uniform curvature. Note that a perfect anti-correlated Pareto front would be linear, which would cause linear scalarizations to always optimize at the end points. We start with simple $k = 2$ case and let $\theta_0$ be random and $\theta_1 = -\theta_0 + \eta$, where $\eta$ is some small random Gaussian perturbation (we set the standard deviation to be about 0.1 times the norm of $\theta_i$). We renormalize after the anti-correlation to ensure $\|\Theta^\star\| = 1$. We run our algorithm with inherent dimension $d = 4$ for $T = 100, 200$ rounds with $k = 2, 6, 10$.

As expected, we find the hypervolume scalarization consistently outperforms the Chebyshev and linear scalarizations, with linear scalarization as the worst performing (see Figure 4). Note that when we increase the output dimension of the problem by setting $k = 10$, the hypervolume scalarization shows a more distinct advantage. The boxed distribution approach of [Paria et al., 2018] does not seem to fare well and consistently performs worse than its uniform counterpart. While linear scalarization provides relatively good performance when the number of objective $k \leq 5$, it appears that as the number of objectives increase in multi-objective optimization, more care needs to be put into the design of scalarization and their weights due to the curse of dimensionality, since the regions of non-uniformity will exponentially increase. We suggest that as more and more objectives are being added to modern machine learning systems, using smart scalarizations is critical to an uniform exploration of the Pareto frontier.

## 5.3 BBOB Functions

We empirically demonstrate the competitiveness of hypervolume scalarizations for Bayesian Optimization by comparing them to the popular BO method of EHVI [Hupkens et al., 2015]. Running our proposed multiobjective algorithms on the Black-Box Optimization Benchmark (BBOB) functions,

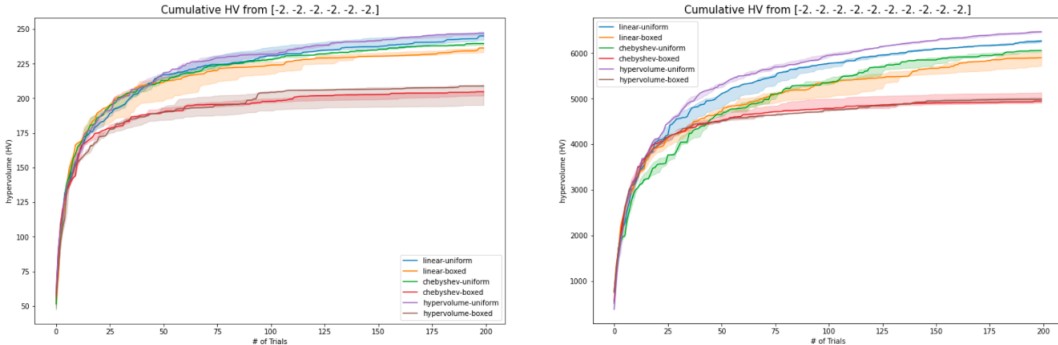

Figure 4: Comparisons of the cumulative hypervolume plots with some anti-correlated $\theta$. When the output dimension increase, there is a clearer advantage to using the hypervolume scalarization over the linear and Chebyshev scalarization. We find that the boxed weight distribution does consistently worse than the uniform distribution.

which can be paired up into multiple bi-objective optimization problems [Tušar et al., 2016]. Our goal is to use a wide set of non-convex benchmarks to supplement our experiments on our simple toy example of linear bandits. For scalarized approaches, we use hypervolume scalarizations with the scalarized UCB algorithm with a constant standard deviation multiplier of $1.8$ and all algorithms with use a Gaussian Process as the underlying model with a standard Matérn kernel that is tuned via ARD Williams and Rasmussen [2006].

Our objectives are given by BBOB functions, which are usually non-negative and are minimized. The input space is always a compact hypercube $[-5, 5]^n$ and the global minima is often at the origin. For bi-objective optimization, given two different BBOB functions $f_1, f_2$, we attempt to maximize the hypervolume spanned by $(-f_1(x_i), -f_2(x_i))$ over choices of inputs $x_i$ with respect to the reference point $(-5, -5)$. We normalize each function due to the drastically different ranges and add random observation noise. We also apply vectorized shifts of the input space of $\mathbf{2}, -\mathbf{2}$ on the two functions respectively, so that the optima of each function do not co-locate at the origin.

We run each of our algorithms in dimensions $d = 8, 16, 24$ and optimize for $160$ iterations with $5$ repeats. From our results, we see that both EHVI and UCB with hypervolume scalarizations are competitive on the BBOB problems but the scalarized UCB algorithm seems to be able to explore the extreme ends of the Pareto frontier, whereas EHVI tends to favor points in the middle (see Appendix and Figure 5). From our experiments, this trend appears to be consistent across different functions and is more prominent as the input dimensions $d$ increase.

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

# A  Linear Bandit Setup and Algorithm

For theory, we use the classic *stochastic linear bandit* setting. For the single-objective setting, in round $t = 1, 2, ..., T$, the and receives a reward $y_t = \langle \theta^\star, a_t \rangle + \xi_t$ where $\xi_t$ is i.i.d. 1-sub-Gaussian noise and $\theta^\star \in \mathbb{R}^d$ is the unknown true parameter vector. In the *multi-objective stochastic linear bandit* setting, the learner chooses an action $a_t \in \mathbb{R}^d$ from the action set $\mathcal{A}$ and receives a vectorized reward $y_t = \Theta^\star a_t + \xi_t$, where $\Theta^\star \in \mathbb{R}^{k \times d}$ is now a matrix of $k$ unknown true parameters and $\xi_t \in \mathbb{R}^k$ is a vector of independent 1-sub-Gaussian noise. We assume, for sake of normalization, that $\|\Theta_i^\star\| \leq 1$ and that $\|a_t\| \leq 1$, where $\|\cdot\|$ denotes the $\ell_2$ norm unless otherwise stated. Other norms that are used include the classical $\ell_p$ norms $\|\cdot\|_p$ and matrix norms $\|x\|_{\mathbf{M}} = x^\top \mathbf{M} x$ for a positive semi-definite matrix $\mathbf{M}$.

We also denote $\mathbf{A}_t \in \mathbb{R}^{d \times t}$ to be the history action matrix, whose $i$-th column is $a_i$, the action taken in round $i$. Similarly, $\mathbf{y}_t$ is defined analogously. Finally, for sake of analysis, we assume that $\mathcal{A}$ contains an isotropic set of actions and specifically, there is $\mathcal{E} \subset \mathcal{A}$ with size $|\mathcal{E}| = O(d)$ such that $\sum_i e_i e_i^\top \succeq \frac{1}{2} \mathbf{I}$, where $\succeq$ denotes the PSD ordering on symmetric matrices. This assumption is not restrictive, as it can be relaxed by using optimal design for least squares estimators [Lattimore and Szepesvári, 2020] and the Kiefer-Wolfrowitz Theorem [Kiefer and Wolfowitz, 1960], which guarantees the existence and construction of an uniform exploration basis of size $O(\text{poly}(d))$.

---

**Algorithm 1:** EXPLOREUCB$(T, \mathcal{D}, s_\lambda)$

---

**Input :** number of maximum actions $T$, weight distribution $\mathcal{D}$ , scalarization $s_\lambda$

1  **repeat**
2  |    Play action $e_n \in \mathcal{E}$ for $n \equiv i \mod d$
3  |    Let $C_{ti}$ be the confidence ellipsoid for $\Theta_i$ and let $UCB_i(a) = \max_{\theta \in C_i} \theta^\top a$
4  |    Draw $\lambda \sim \mathcal{D}$, play $a^* = \text{argmax}_{a \in \mathcal{A}} s_\lambda(UCB_i(a))$
5  **until** *number of rounds $i$ exceed $T/2$*

---

# B  Missing Proofs

*Proof of Lemma 5.* Let $\lambda = y^\star / \|y^\star\|$. Note that $\lambda > 0$ since $y^\star$ is in the positive orthant and for the sake of contradiction, assume there exists $z$ such that $s_\lambda(z) > s_\lambda(y^\star)$. However, note that for any $i$, $\frac{z_i}{\lambda_i} \geq \min_i \frac{z_i}{\lambda_i} > \min_i \frac{y_i^\star}{\lambda_i} = \frac{y_i^\star}{\lambda_i}$, where the last line follows since $y_i^\star / \lambda_i = \|y^\star\|$ for all $i$ by construction. Therefore, we conclude that $y^\star < z$, contradicting that $y^\star$ is Pareto optimal.

Finally, note that if $\alpha\lambda$ is on the Pareto frontier, then we see that $\min_i \alpha\lambda_i / \lambda_i = \alpha$ and furthermore, this min value is achieved for all $i$. Therefore, since $\alpha\lambda$ is on the Pareto frontier, any other point $y \in \mathcal{Y}$ has some coordinate $j$ such that $y_j < \alpha\lambda_j$, which implies that $\min_i y_i / \lambda_i < \alpha$. $\qquad\square$

*Proof of Theorem 7.* Let us denote $s_\lambda = s_\lambda^{HV}$ is the hypervolume scalarization and WLOG let $z = 0$. Note that we can first decompose our regret as

$$\mathcal{HV}_z(\mathcal{Y}^\star) - \mathcal{HV}_z(\mathcal{Y}_T) \leq |\mathcal{HV}_z(\mathcal{Y}^\star) - \frac{c_k}{T} \sum_{i=1}^T \max_{y \in \mathcal{Y}} s_{\lambda_i}(y)| + |\frac{c_k}{T} \sum_{i=1}^T \max_{y \in \mathcal{Y}} s_{\lambda_i}(y) - \mathcal{HV}_z(\mathcal{Y}_T)|$$

$$\leq |\mathcal{HV}_z(\mathcal{Y}) - \frac{c_k}{T} \sum_{i=1}^T s_{\lambda_i}(y)| + |\frac{c_k}{T} \sum_{i=1}^T \max_{y \in \mathcal{Y}_T} s_{\lambda_i}(y) - \mathcal{HV}_z(\mathcal{Y}_T)|$$

where the second inequality uses the fact that $y_i \in \arg\max_{y \in \mathcal{Y}} s_{\lambda_i}(y)$. We proceed to bound both parts separately and we note that it suffices to prove uniform concentration of the empirical sum to the expectation, as by our choice of scalarization is the hypervolume by Lemma 6.

Let $f_{\mathbf{Y}}(\lambda_i) = \max_{y \in \mathbf{Y}} s_{\lambda_i}(y)$. We let $\mathcal{F} = \{f_{\mathbf{Y}} : \mathbf{Y} \subseteq \mathcal{A}\}$ be our class of functions over all possible output sets $\mathbf{Y}$. We will first demonstrate uniform convergence by bounding the complexity of $\mathcal{F}$. Specifically, by generalization bounds from Rademacher complexities Bartlett and Mendelson [2002], over choices of $\lambda_i \sim \mathcal{D}$, we know that with probability $1 - \delta$, for all $\mathbf{Y}$, we have the bound

$$\left| \mathop{\mathbf{E}}_{\lambda \sim \mathcal{D}}[f_{\mathbf{Y}}] - \frac{1}{m} \sum_{i=1}^{m} f_{\mathbf{Y}}(\lambda_i) \right| \leq R_m(\mathcal{F}) + \sqrt{\frac{8 \ln(2/\delta)}{m}}$$

where $R_m(\mathcal{F}) = \mathbf{E}_{\lambda_i \sim \mathcal{D}, \sigma_i} \left[ \sup_{f \in \mathcal{F}} \frac{2}{m} \sum_i \sigma_i f(\lambda_i) \right]$, where $\sigma_i$ are i.i.d. $\pm 1$ Rademacher variables.

To bound $R_m(\mathcal{F})$, we appeal to Dudley's integral formulation that allows us to use the metric entropy of $\mathcal{F}$ to bound

$$R_m(\mathcal{F}) \leq \inf_{\alpha > 0} \left( 4\alpha + 12 \int_{\alpha}^{\infty} \sqrt{\frac{\log(\mathcal{N}(\epsilon, \mathcal{F}, \|\cdot\|_2))}{m}} \, d\epsilon \right)$$

where $\mathcal{N}$ denotes the standard covering number for $\mathcal{F}$ under the $\ell_2$ function norm metric over $\lambda \in \mathcal{D}$.

Since $\mathcal{D}$ is the uniform distribution over $\mathcal{S}_+$, this induces a natural $\ell_\infty$ function norm metric on $\mathcal{F}$ that is $\|f\|_\infty = \sup_{\lambda \in \mathcal{S}_+} |f(\lambda)|$. By Lemma 15, since $y_i - z$ is bounded below and above by $B_u, B_l$ respectively, $s_\lambda(y)$ is $L_\lambda$ Lipschitz with respect to the Euclidean norm in $\lambda$. Note that since the maximal operator preserves Lipschitzness, $f_{\mathbf{Y}}(\lambda)$ is also $L_\lambda$-Lipschitz with respect to $\lambda \in \mathbb{R}^k$ for any $\mathbf{Y}$. Since $\mathcal{F}$ contains $L_\lambda$-Lipschitz functions in $\mathbb{R}^k$, we can bound the metric entropy via a covering of $\lambda$ via a Lipschitz covering argument (see Lemma 4.2 of Gottlieb et al. [2016]), so we have

$$\log(\mathcal{N}(\epsilon, \mathcal{F}, \|\cdot\|_2)) \leq \log(\mathcal{N}(\epsilon, \mathcal{F}, \|\cdot\|_\infty)) \leq (4L_\lambda/\epsilon)^k \log(8/k)$$

Finally, we follow the same Dudley integral computation of Theorem 4.3 of Gottlieb et al. [2016] to get that

$$R_m(\mathcal{F}) \leq \inf_{\alpha > 0} \left( 4\alpha + 12 \int_{\alpha}^{2} \sqrt{\frac{(4L_\lambda/\epsilon)^k \log(8/k)}{T}} \, d\epsilon \right)$$

$$= O(L_\lambda / m^{1/(k+1)})$$

Therefore, we conclude that with probability at least $1 - \delta$ over the independent choices of $\lambda_i \sim \mathcal{D}$, for all $\mathbf{Y}$ and setting $m = T$

$$\left| \mathop{\mathbf{E}}_{\lambda \sim \mathcal{D}} \left[ \max_{a \in \mathbf{Y}} s_\lambda(\Theta^\star a) \right] - \frac{1}{T} \sum_{i=1}^{T} \max_{a \in \mathbf{Y}} s_{\lambda_i}(\Theta^\star a) \right|$$

$$\leq O\left( \frac{L_\lambda}{T^{1/(k+1)}} \right) + \sqrt{\frac{8 \ln(2/\delta)}{T}}$$

where $B$ is some $\ell_\infty$ upper bound on the output set $\mathbf{Y}$. Finally, we conclude by using Lemma 6 to replace the expectation by the hypervolume and by setting $\mathbf{Y} = \mathcal{Y}, \mathcal{Y}_T$ respectively.

Now consider the case when $k = 2$ and we are using the Chebyshev scalarizer $s_\lambda^{CS}$. We claim that the distribution $\mathbf{Y}_T = \{y_1, ..., y_T\}$ is the same and specifically let $Y$ denote that random variable over the choice of $\lambda \sim \mathcal{S}_+$ that corresponds to $y \in \arg\max_{y \in \mathcal{Y}} s_\lambda^{CS}(y)$. Then, note that a standard way to draw a random weight on $\mathcal{S}_+$ is to draw random absolute Gaussians and then renormalize, so if $R^2 = \sum_i |X_i|^2$, where $X_i$ are i.i.d. Gaussian, then $Y = \arg\max_{y \in \mathcal{Y}} \min(\frac{|X_1|}{R} y_1, \frac{|X_2|}{R} y_2)$. Note that the optimization of the arg-max is unaffected by positive scalar multiples of the objective, so multiplying by $R^2/(|X_1||X_2|)$ gives $Y = \arg\max_{y \in \mathcal{Y}} \min(\frac{R}{|X_2|} y_1, \frac{R}{|X_1|} y_2)$. Note since $X_1, X_2$ are i.i.d., we conclude that $Y$ is the same distribution as the random variable that is given by

$y \in \arg\max_{y \in \mathcal{Y}} \sqrt{s_\lambda^{\text{HV}}(y)}$. Since monotone transforms of the objective does not change the arg-max distribution, we conclude.

$\square$

## B.1 Proofs of Lipschitz Properties

Again, recall that for a scalarization function $s_\lambda(x)$, $s_\lambda$ is $L_p$-Lipschitz with respect to $x$ in the $\ell_p$ norm on $\mathcal{X}$ if for $x_1, x_2 \in \mathcal{X}$, $|s_\lambda(x_1) - s_\lambda(x_2)| \leq L_p\|x_1 - x_2\|_p$, and analogously for define $L_\lambda$ for $p = 2$ so $|s_{\lambda_1}(x) - s_{\lambda_2}(x)| \leq L_\lambda\|\lambda_1 - \lambda_2\|_2$. We utilize the fact that if $s_\lambda$ is differentiable everywhere except for a finite set, bounding Lipschitz constants is equivalent to bounding the dual norm $\|\nabla s_\lambda\|_q$, where $1/p + 1/q = 1$, which follows from mean value theorem, which we state as Proposition 12.

**Proposition 12.** *Let $f : \mathcal{X} \to \mathbb{R}$ be a continuous function that is differentiable everywhere except on a finite set, then if $\|\nabla f(x)\|_q \leq L_p$ for all $x \in \mathcal{X}$, $f(x)$ is $L_p$-Lipschitz with respect to the $\ell_p$ norm.*

**Lemma 13.** *Let $s_\lambda(y) = \lambda^\top y$ be the linear scalarization with $\|\lambda\| \leq 1$ and $\|y\|_\infty \leq 1$. Then, we may bound $L_p \leq \max(1, k^{1/2-1/p})$ and $L_\lambda \leq \sqrt{k}$ and $|s_\lambda| \leq \sqrt{k}$.*

*Proof of Lemma 13.* Since $\nabla_\lambda s_\lambda(x) = y$, we use Proposition 12 to bound $L_\lambda \leq \max_y \|y\| \leq \sqrt{k}\|y\|_\infty = \sqrt{k}$. Similarly, since $\nabla_y s_\lambda(y) = \lambda$, we may bound for $p \leq 2$, $L_p \leq \|\lambda\|_q \leq \|\lambda\| \leq 1$ for $1/p + 1/q = 1$ and for $p \geq 2$, we may use Holder's inequality to bound $L_p \leq \|\lambda\|_q \leq k^{1/q-1/2}\|\lambda\| \leq k^{1/2-1/p}$.

To bound the absolute value of $s_\lambda$, note $s_\lambda(y) = \lambda^\top y \leq \sqrt{k}$ for all since $\|y\|_2 \leq \sqrt{k}\|y\|_\infty \leq \sqrt{k}$.

$\square$

**Lemma 14.** *Let $s_\lambda(y) = \min_i \lambda_i y_i$ be the Chebyshev scalarization with $\|\lambda\| \leq 1$ and $\|y\|_\infty \leq 1$. Then, we may bound $L_p \leq 1$ and $L_\lambda \leq 1$ and $|s_\lambda| \leq 1/\sqrt{k}$.*

*Proof of Lemma 14.* For a specific $\lambda, y$, let $i^\star$ be the optimal index of the minimization. Then, the gradient $\nabla_\lambda s_\lambda(x)$ is simply zero in every coordinate except at $i^\star$, where it is $y_{i^\star}$. Therefore, since we can only have a finite number of discontinuities due to monotonicity, we use Proposition 12 to bound $L_\lambda \leq y_{i^\star} \leq 1$. Similarly, since $\nabla_y s_\lambda(y)$ has only one non-zero coordinate except at $i^\star$, which is $\lambda_{i^\star}$, we may bound for $L_q \leq \lambda_{i^\star} \leq 1$.

To bound the absolute value of $s_\lambda$, note that there must exists $\lambda_i < 1/\sqrt{k}$ as $\|\lambda\| \leq 1$. Thus, $\min_i \lambda_i y_i < 1/\sqrt{k}$ for $\|y\|_\infty \leq 1$.

$\square$

**Lemma 15.** *Let $s_\lambda(y) = \min_i (y_i/\lambda_i)^k$ be the hypervolume scalarization with $\|\lambda\| = 1$ and $0 < B_l \leq y_i \leq B_u$. Then, we may bound $L_p \leq \frac{B_u^k}{B_l k^{k/2-1}}$ and $L_\lambda \leq \frac{B_u^{k+1}}{B_l k^{(k-1)/2}}$ and $|s_\lambda| \leq \frac{B_u^k}{k^{k/2}}$.*

*Proof of Lemma 15.* For a specific $\lambda, y$, let $i^\star$ be the optimal index of the minimization. Then, the gradient $\nabla_\lambda s_\lambda(x)$ is simply zero in every coordinate except at $i^\star$, which in absolute value is $k(y_{i^\star}/\lambda_{i^\star})^k(1/\lambda_{i^\star})$.

Let $j$ be the index such that $\lambda_j$ is maximized and since $\|\lambda\| = 1$, we know that $\lambda_j \geq 1/\sqrt{k}$. Therefore, we see that since $y_{i^\star}/\lambda_{i^\star} \leq y_j/\lambda_j \leq y_j/\sqrt{k}$, we conclude that $1/\lambda_{i^\star} \leq (B_u/B_l)/\sqrt{k}$.

Therefore, using Proposition 12, we have $L_\lambda \leq k(y_{i^\star}/\lambda_{i^\star})^k(1/\lambda_{i^\star}) \leq k(B_u/\sqrt{k})^k \frac{(B_u/B_l)}{\sqrt{k}} = \frac{B_u^{k+1}}{B_l k^{(k-1)/2}}$

And similarly, since $\nabla_y s_\lambda(y)$ has only one non-zero coordinate except at $i^\star$, which is $k(y_{i^\star}/\lambda_{i^\star})^{k-1}(1/\lambda_{i^\star})$, we may bound for

$$L_q \leq k(y_{i^\star}/\lambda_{i^\star})^{k-1}(1/\lambda_{i^\star}) \leq k(B_u/\sqrt{k})^{k-1}\frac{(B_u/B_l)}{\sqrt{k}} \leq \frac{B_u^k}{B_l k^{k/2-1}}$$

To bound the absolute value of $s_\lambda$, note that $s_\lambda(y) \leq (\frac{y_j}{\lambda_j})^k \leq \frac{B_u^k}{k^{k/2}}$.

$\square$

## B.2 Proofs for Linear Bandits

The following lemma about the UCB ellipsoid is borrowed from the original analysis of linear bandits.

**Lemma 16** (Abbasi-Yadkori et al. [2011]). *Consider the least squares estimator $\widehat{\theta}_t = (\mathbf{M}_t)^{-1}\mathbf{A}_t^\top \mathbf{y}_t$, where the covariance matrix of the action matrix is $\mathbf{M}_t = \mathbf{A}_t^\top \mathbf{A}_t + \lambda\mathbf{I}$, then with probability $1 - \delta$,*

$$\|\widehat{\theta}_t - \theta^*\|_{\mathbf{M}_t} \leq \sqrt{\lambda}\|\theta^*\| + \sqrt{2\log(\frac{1}{\delta}) + d\log(T/\lambda)}$$

*Proof of Lemma 10.* Let $\widehat{\Theta}_T$ be the least squares estimate of the true parameters after observing $(\mathbf{A}_T, \mathbf{y}_T)$. Since the noise $\xi_t$ in each objective is independent and 1-sub-Gaussian, by Lemma 16, if we let $\mathbf{M}_T = \mathbf{A}_T^\top\mathbf{A}_T + \lambda\mathbf{I}$, then with regularization $\lambda = 1$

$$\|\widehat{\Theta}_{Ti} - \Theta_i^\star\|_{\mathbf{M}_T} \leq 1 + \sqrt{2\log(k/\delta) + d\log(T)} := D_T$$

holds with probability at least $1 - \delta/k$. Note that this describes the confidence ellipsoid, $C_{Ti} = \{\theta \in \mathbb{R}^d : \|\widehat{\Theta}_{Ti} - \theta_i\|_{\mathbf{M}_T} \leq D_T\}$ for $\Theta_{Ti}$.

By the definition of the UCB maximization of $a_t$, we see $a_t, \widetilde{\Theta}_t = \arg\max_{a\in\mathcal{A}} \max_{\theta_i \in C_{Ti}} s_\lambda(\Theta_i^\top a)$. Note that since $\Theta^\star \in \mathbf{C}_T$, we can bound the instantaneous scalarized regret as:

$$r(s_\lambda, a_t) = \max_{a\in\mathcal{A}} s_\lambda(\Theta^\star a) - s_\lambda(\Theta^\star a_t) \leq s_\lambda(\widetilde{\Theta}_t a_t) - s_\lambda(\Theta^\star a_t)$$

By the Lipschitz smoothness condition, we conclude that $r(s_\lambda, a_t) \leq L_p\|(\widetilde{\Theta}_t - \Theta^\star)a_t\|_p$.

To bound the desired $\ell_p$ norm, first note that by triangle inequality, $\|\widetilde{\Theta}_t - \Theta^\star\|_{\mathbf{M}_T} \leq 2D_T$. Since we apply uniform exploration every other step and $\sum_i e_i e_i^\top \succeq \frac{1}{2}\mathbf{I}$ for $e_i \in \mathcal{E}$ with size $|\mathcal{E}| = d$, we conclude that $\mathbf{M}_T \succeq \frac{T}{5d}\mathbf{I}$. Therefore, we conclude that $\|\widehat{\Theta}_{Ti} - \Theta_i^\star\| \leq 5\sqrt{d/T}D_T := E_T$ with probability at least $1 - \delta/k$. Since $\|a_t\| \leq 1$, we conclude by Cauchy-Schwarz, that $|(\widehat{\Theta}_{Ti} - \Theta_i^\star)a_t| \leq E_T$. Together with our Lipschitz condition, we conclude that

$$r(s_\lambda, a_t) \leq k^{1/p}L_p E_T \leq 10k^{1/p}L_p d\sqrt{(\log(k/\delta) + \log(T))/T}$$

. $\qquad\qquad\qquad\qquad\qquad\qquad\qquad\qquad\qquad\qquad\qquad\qquad\qquad\qquad\qquad\square$

**Theorem 17.** *Assume that for any $a \in \mathcal{A}$, $|s_\lambda(\Theta^\star a)| \leq B$ for some $B$ and $s_\lambda$ is $L_\lambda$-Lipschitz with respect to the $\ell_2$ norm in $\lambda$. With constant probability, the Bayes regret of running Algorithm 1 at round $T$ can be bounded by*

$$BR(s_\lambda, \mathbf{A}_T) \leq O\left(k^{\frac{1}{p}}L_p d\sqrt{\frac{\log(kT)}{T}} + \frac{BL_\lambda}{T^{\frac{1}{k+1}}}\right)$$

*Proof of Theorem 17.* For any set of actions $\mathbf{A} \subseteq \mathcal{A}$, we define $f_{\mathbf{A}}(\lambda) = \max_{a\in\mathbf{A}} s_\lambda(\Theta^\star a)$. We let $\mathcal{F} = \{f_{\mathbf{A}} : \mathbf{A} \subseteq \mathcal{A}\}$ be our class of functions over all possible action sets and for any Bayes regret bounds, we will first demonstrate uniform convergence by bounding the complexity of $\mathcal{F}$. Specifically, by generalization bounds from Rademacher complexities Bartlett and Mendelson [2002], over choices of $\lambda_i \sim \mathcal{D}$, we know that with probability $1 - \delta$, for all $\mathbf{A}$, we have the bound

$$\left|\mathop{\mathbf{E}}_{\lambda\sim\mathcal{D}}[f_{\mathbf{A}}] - \frac{1}{m}\sum_{i=1}^m f_{\mathbf{A}}(\lambda_i)\right| \leq R_m(\mathcal{F}) + \sqrt{\frac{8\ln(2/\delta)}{m}}$$

where $R_m(\mathcal{F}) = \mathbf{E}_{\lambda_i\sim\mathcal{D},\sigma_i}\left[\sup_{f\in\mathcal{F}}\frac{2}{m}\sum_i \sigma_i f(\lambda_i)\right]$, where $\sigma_i$ are i.i.d. $\pm 1$ Rademacher variables.

To bound $R_m(\mathcal{F})$, we appeal to Dudley's integral formulation that allows us to use the metric entropy of $\mathcal{F}$ to bound

$$R_m(\mathcal{F}) \leq \inf_{\alpha>0} \left( 4\alpha + 12 \int_\alpha^\infty \sqrt{\frac{\log(\mathcal{N}(\epsilon, \mathcal{F}, \|\cdot\|_2))}{m}} \, d\epsilon \right)$$

where $\mathcal{N}$ denotes the standard covering number for $\mathcal{F}$ under the $\ell_2$ function norm metric over $\lambda \in \mathcal{D}$.

Since $\mathcal{D}$ is the uniform distribution over $\mathcal{S}_+$, this induces a natural $\ell_\infty$ function norm metric on $\mathcal{F}$ that is $\|f\|_\infty = \sup_{\lambda \in \mathcal{S}_+} |f(\lambda)|$. Since $s_\lambda(\Theta^\star a)$ is $L_\lambda$ Lipschitz with respect to the Euclidean norm in $\lambda$. Note that since the maximal operator preserves Lipschitzness, $f_{\mathbf{A}}(\lambda)$ is also $L_\lambda$-Lipschitz with respect to $\lambda \in \mathbb{R}^k$. Since $\mathcal{F}$ contains $L_\lambda$-Lipschitz functions in $\mathbb{R}^k$, we can bound the metric entropy via a covering of $\lambda$ via a Lipschitz covering argument (see Lemma 4.2 of Gottlieb et al. [2016]), so we have

$$\log(\mathcal{N}(\epsilon, \mathcal{F}, \|\cdot\|_2)) \leq \log(\mathcal{N}(\epsilon, \mathcal{F}, \|\cdot\|_\infty)) \leq (4L_\lambda/\epsilon)^k \log(8/k)$$

Finally, we follow the same Dudley integral computation of Theorem 4.3 of Gottlieb et al. [2016] to get that

$$R_m(\mathcal{F}) \leq \inf_{\alpha>0} \left( 4\alpha + 12 \int_\alpha^2 \sqrt{\frac{(4L_\lambda/\epsilon)^k \log(8/k)}{m}} \, d\epsilon \right)$$

$$= O(L_\lambda/m^{1/(k+1)})$$

Therefore, we conclude that with probability at least $1 - \delta$ over the independent choices of $\lambda_i \sim \mathcal{D}$, for all $\mathbf{A}$,

$$\left| \mathop{\mathbf{E}}_{\lambda \sim \mathcal{D}} \left[ \max_{a \in \mathbf{A}} s_\lambda(\Theta^\star a) \right] - \frac{1}{m} \sum_{i=1}^m \max_{a \in \mathbf{A}} s_{\lambda_i}(\Theta^\star a) \right|$$

$$\leq O\left( \frac{BL_\lambda}{m^{1/(k+1)}} \right) + \sqrt{\frac{8\ln(2/\delta)}{m}}$$

Finally, note that for $T$ even, with constant probability,

$$BR(s_\lambda, \mathbf{A}_t) = \mathop{\mathbf{E}}_{\lambda \sim \mathcal{D}}[r(s_\lambda, \mathbf{A}_t)]$$

$$= \mathop{\mathbf{E}}_{\lambda \sim \mathcal{D}}[\max_{a \in \mathcal{A}} s_\lambda(\Theta^\star a) - \max_{a \in \mathbf{A}_T} s_\lambda(\Theta^\star a)]$$

$$\leq \frac{1}{T/2} \sum_{i=1}^{T/2} \left[ \max_{a \in \mathcal{A}} s_{\lambda_i}(\Theta^\star a) - \max_{a \in \mathbf{A}_T} s_{\lambda_i}(\Theta^\star a) \right]$$

$$+ O\left( \frac{BL_\lambda}{T^{1/(k+1)}} \right)$$

$$\leq \frac{1}{T/2} \sum_{i=1}^{T/2} \left[ \max_{a \in \mathcal{A}} s_{\lambda_i}(\Theta^\star a) - s_{\lambda_i}(\Theta^\star a_{2i}) \right]$$

$$+ O\left( \frac{BL_\lambda}{T^{1/(k+1)}} \right)$$

$$\leq \frac{1}{T/2} \sum_{i=1}^{T/2} r(s_{\lambda_i}, a_{2i}) + O\left( \frac{BL_\lambda}{T^{1/(k+1)}} \right)$$

$$\leq O\left( k^{1/p} L_p d \sqrt{\frac{\log(kT)}{T}} + \frac{BL_\lambda}{T^{1/(k+1)}} \right)$$

where the last line used Lemma 10 with $\delta = 1/T^2$ and applied a union bound over all $O(T)$ iterations. □

*Proof of Theorem 11.* Note that by Lemma 6, we connect the Bayes regret to the hypervolume regret for $\mathcal{D}$ :

$$\mathcal{HV}_z(\Theta^\star \mathcal{A}) - \mathcal{HV}_z(\Theta^\star \mathbf{A}_t)$$

$$= c_k \underset{\lambda \sim \mathcal{D}}{\mathbf{E}} [\max_{a \in \mathcal{A}} s_\lambda(\Theta^\star a) - \max_{a \in \mathbf{A}_T} s_\lambda(\Theta^\star a)]$$

where $s_\lambda(y) = \min_i (y_i - z_i/\lambda_i)^k$.

Note that since $\|\Theta^\star a\|_\infty \leq 1$ for all $a \in \mathcal{A}$ and $B$ is maximal, we have $B \leq \Theta^\star a - z \leq B + 2$. Therefore, we conclude by Lemma 15 that $s_\lambda$ is Lipschitz with

$$L_p \leq \frac{(B+2)^k}{Bk^{k/2-1}}, L_\lambda \leq \frac{(B+2)^{k+1}}{Bk^{(k-1)/2}}, |s_\lambda| \leq \frac{(B+2)^k}{k^{k/2}}$$

Finally, we combine this with Theorem 17 with $p = \infty$ as the optimal choice of $p$ (since $L_p$ does not depend on $p$) to get our desired bound on hypervolume regret. □

*Proof of Theorem 8.* We let $\mathcal{A} = \{a : \|a\| \leq 1\}$ be the unit sphere and $\Theta_i^\star = e_i$ be the unit vector directions. Note that in this case the Pareto frontier is exactly $\mathcal{S}_+^{k-1}$.

Consider a uniform discretization of the Pareto front by taking an $\epsilon$ grid with respect to each angular component with respect to the polar coordinates. Let $p_1, ..., p_m$ be the center (in terms of each of the $k-1$ angular dimensions) in the $m = \Theta((1/\epsilon)^{k-1})$ grid elements. We consider the output $\mathbf{y}_T = \Theta^\star \mathbf{A}_T$ and assume that for some grid element $i$, it contains none of the $T$ outputs $\mathbf{y}_T$. Since our radial component $r = 1$, by construction of our grid in the angular component, we deduce that $\min_t \|y_t - p_i\|_\infty > \epsilon/10$ by translating polar to axis-aligned coordinates.

Let $\epsilon' = \epsilon/10$. Assume also that $\frac{1}{k} < p_i < 1 - \frac{1}{k}$. Next, we claim that the hypercube from $p_i - \epsilon'/k^2$ to $p_i$ is not dominated by any points in $\mathbf{Y}_T$. Assume otherwise that there exists $y_t$ such that $y_t \geq p_i - \epsilon'/k^2$. Now, this combined with the fact that since $\min_t \|y_t - p_i\|_\infty > \epsilon'$ implies that there must exist a coordinate such that $y_{tj} \geq p_j + \epsilon'$.

However, this implies that

$$\sum_{i=1}^k y_{ti}^2 \geq \sum_{i \neq j} (p_i - \epsilon'/k^2)^2 + (p_j + \epsilon')^2$$

$$\geq \sum_i p_i^2 - 2(\epsilon'/k^2) \sum_{i \neq j} p_i + 2\epsilon' p_j > 1$$

where the last inequality follows since $\sum_i p_i < 1/\sqrt{k}$ and $p_j > \frac{1}{k}$ by assumption. However, this contradicts that $\|y_t\| \leq 1$, so it follows that $p_i - \epsilon'$ is not dominated.

Therefore, for any grid element such that $p_i > 1/k$, if there is no $y_t$ in the grid, we must have a hypervolume regret of at least $\Omega(\epsilon'^k) = \Omega(\epsilon^k)$ be simply consider the undominated hypervolume from $p_i$ to $p_i - \epsilon'$, which lies entirely within the grid element. In fact, since there are $\Theta((1/\epsilon)^{k-1})$ such grid elements satisfying $p_i > 1/k$, we see that if $T < O((1/\epsilon)^{k-1})$, by pigeonhole, there must be a hypervolume regret of at least $\Omega((1/\epsilon)^{k-1} \epsilon^k) = \Omega(\epsilon)$

Therefore, for any $1/2 > \epsilon > 0$, $\mathcal{HV}_z(\Theta^\star \mathcal{A}) - \mathcal{HV}_z(\Theta^\star \mathbf{A}_T) < \epsilon$ implies that $T = \Omega((1/\epsilon)^{k-1})$. Rearranging shows that

$$\mathcal{HV}_z(\Theta^\star \mathcal{A}) - \mathcal{HV}_z(\Theta^\star \mathbf{A}_T) = \Omega(T^{-1/(k-1)})$$

□

# C Figures

## C.1 Synthetic Figures

Here are the relevant full plots from synthetic optimization of Pareto frontiers. Note that the title of each plots explicitly mention the optimized function. Generally, we observe that the hypervolume scalarizer has better performance on more concave curves. We also include a plot of a concave function multiplied by a linear combination of a convex and concave function, given by $z = \exp(-x)(x \exp(-y) + (1-x)(3 - \exp(y)))$, which demonstrates that hypervolume scalarization performs competitively even with convex frontier regions.

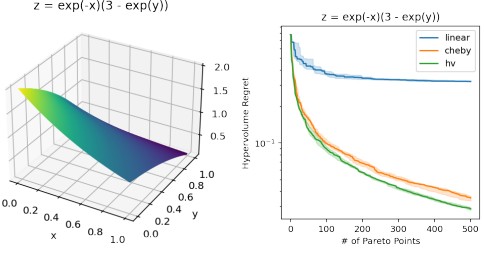

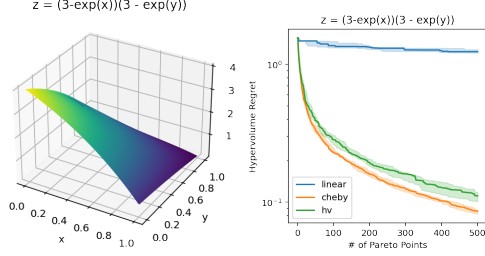

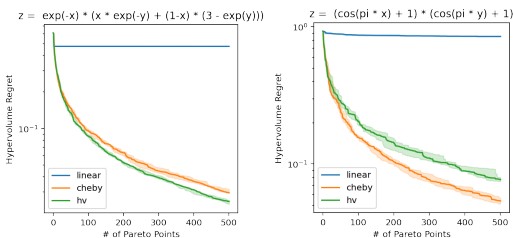

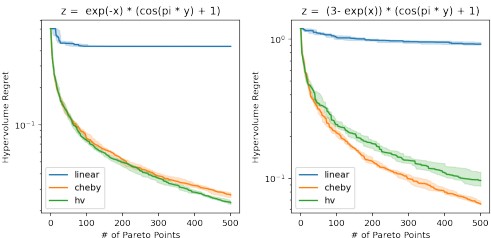

## C.2   BBOB Figures

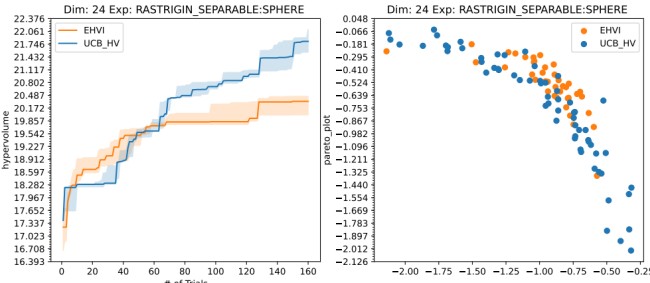

Figure 5: Comparisons of the hypervolume indicator and the optimization fronts with BBOB functions. The left plot tracks the dominated hypervolume as a function of trials that were evaluated. The blue/orange dots represent the frontier points of the UCB-HV/EHVI algorithms respectively, over 5 repeats. Especially in high dimensions, EHVI tends overly concentrate on points in the middle of the frontier, limiting its hypervolume gain, while hypervolume scalarizations produce more diverse points.

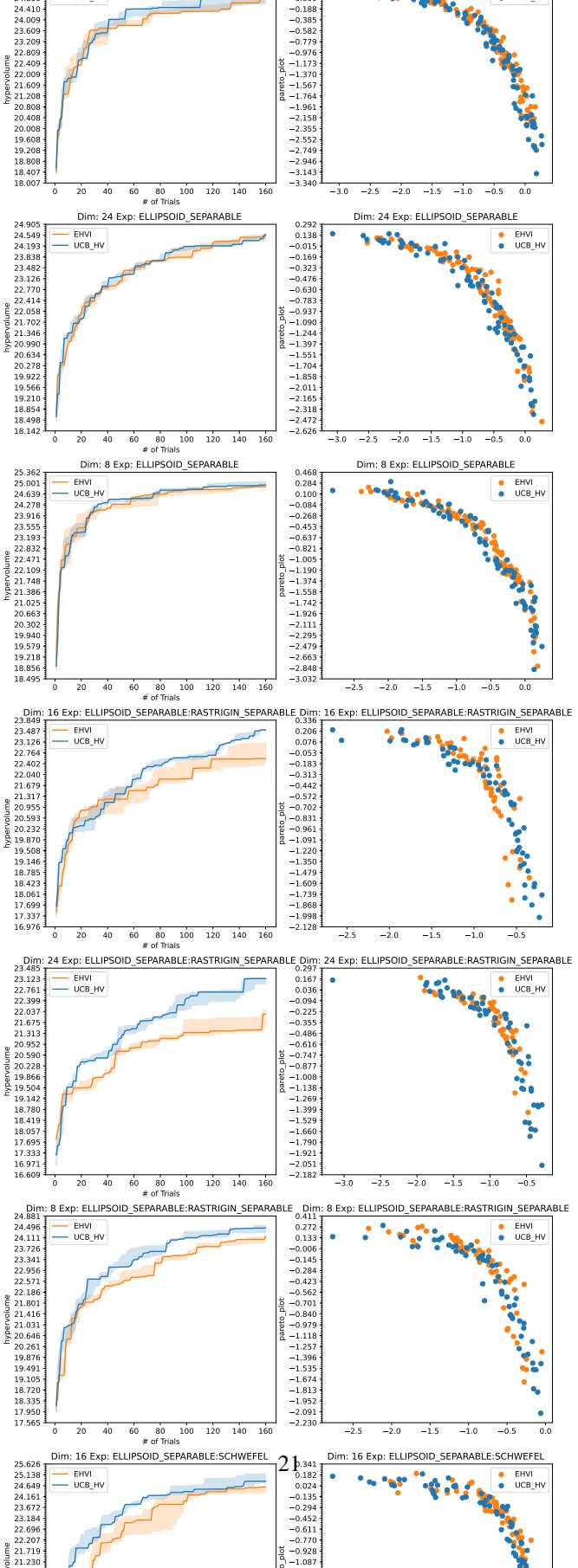

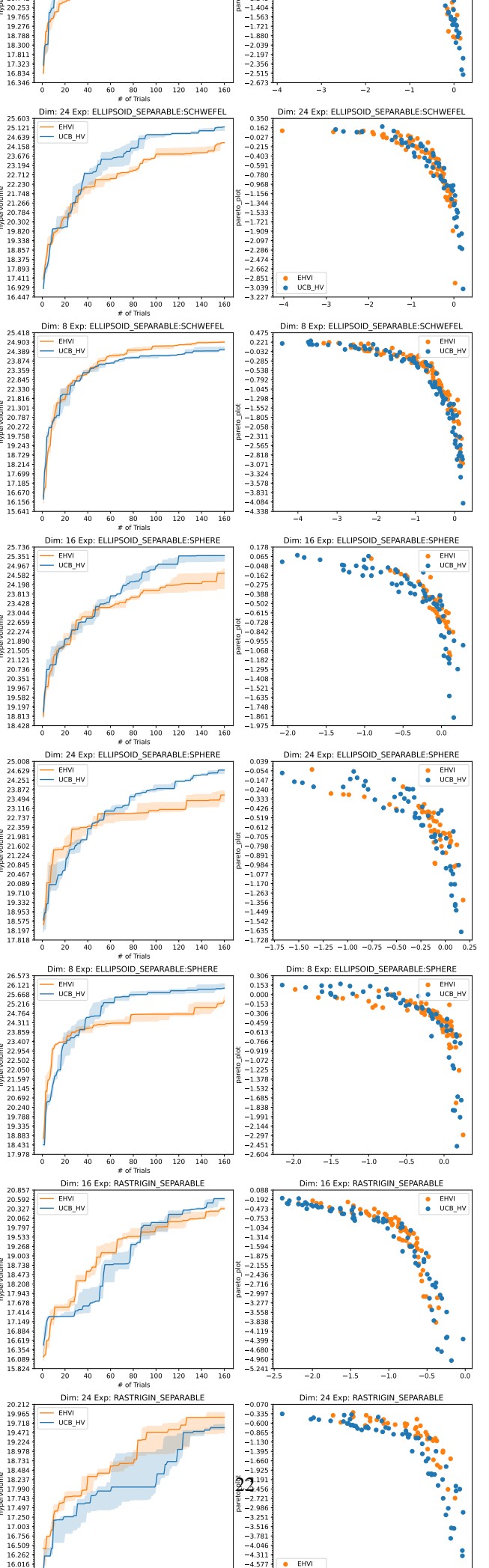

## D   Code

```
# -*- coding: utf-8 -*-
"""3D Hypervolume Experiments

Automatically generated by Colab.

Original file is located at
    https://colab.corp.google.com/drive/1XbxabRf_aqgE0nRY-cX6McieUc3XnHSG

### Imports
"""

import numpy as np
import matplotlib.pyplot as plt
from vizier.pyvizier import multimetric
from vizier.pyvizier.multimetric import xla_pareto

eps = 1e-4
origin = np.zeros(shape=3) - eps
front = multimetric.ParetoFrontier(
        points=np.zeros(shape=(2,3)) - eps,
        origin=origin,
        num_vectors=10000,
        cum_hypervolume_base=xla_pareto.jax_cum_hypervolume_origin,
)
hv_curve = front.hypervolume(is_cumulative=True)

import numpy as np
import matplotlib.pyplot as plt

pi = np.pi + 1e-4
# Define the function to plot
threshold = 0.1
def g(x):
  if x <= threshold:
    return np.cos(x*pi/(2*threshold)) + 1
  else:
    return np.exp(-3*(x-threshold))
g = lambda x : np.cos(x*pi) + 1

# Generate some data
x = np.linspace(0, 1, 500)
y = np.vectorize(g)(x)

# Plot the data
plt.plot(x, y, 'bo')
plt.xlabel('x')
plt.ylabel('y')
plt.title('1D Function Plot')
plt.show()

def f(x, y):
  # return np.vectorize(g)(x) * np.vectorize(g)(y)
 # return (3 - np.exp(x)) * (x * np.exp(-y) + (1-x)*(3-np.exp(y)))
 # return (3 - np.exp(x))*(3- np.exp(y))
```

```python
  # return np.exp(-x) * (3-np.exp(y))
  return (3-np.exp(x)) * (np.cos(y * pi) + 1)
  return np.exp(-x) * (x * np.exp(-y) + (1-x) * (3 - np.exp(y)))
    # return np.exp(-x - y)

x = np.linspace(0, 1, 30)
y = np.linspace(0, 1, 30)

X, Y = np.meshgrid(x, y)
Z = f(X, Y)
fig = plt.figure(figsize=(4, 4))
ax = plt.axes(projection='3d')
ax.contour3D(X, Y, Z, 500)
ax.set_xlabel('x')
ax.set_ylabel('y')
ax.set_zlabel('z')
ax.set_title('z = (3-exp(x))(3 - exp(y)) ')

x = X.flatten()
y = Y.flatten()
z = Z.flatten()

xla_pareto.is_frontier(np.array([x, y, z]).T)

total_hypervolume = front.hypervolume(additional_points=np.array([x, y, z]).T)

total_hypervolume

#@title Hypervolume code
def get_points(scalarizer_generator, num_points = 50):
  index_maxes = []
  for _ in range(num_points):
    outputs = scalarizer_generator(np.array([x, y, z]))
    index_max = np.argmax(outputs)
    index_maxes.append(index_max)
  return index_maxes

def linear_gaussian_generator(array):
  weights = abs(np.random.normal(size=(array.shape[0], 1)))
  return np.sum(weights * array, axis=0)

def cheby_gaussian_generator(array):
  weights = abs(np.random.normal(size=(array.shape[0], 1)))
  return np.min(weights * array, axis=0)

def hv_gaussian_generator(array):
  weights = abs(np.random.normal(size=(array.shape[0], 1)))
  return np.min((1/weights) * array, axis=0)

generators = {
    'linear': linear_gaussian_generator,
    'cheby': cheby_gaussian_generator,
    'hv': hv_gaussian_generator,
}

index_maxes = get_points(hv_gaussian_generator)

from vizier.benchmarks.analyzers import plot_median_convergence
```

```python
num_repeats = 10
generator_curves = {}
for _ in range(num_repeats):
  for key, generator in generators.items():
    index_maxes = get_points(generator, num_points=500)
    new_points = np.vstack([x[index_maxes], y[index_maxes], z[index_maxes]]).T
    hv_curve = front.hypervolume(is_cumulative=True, additional_points=new_points)
    regret_curve = total_hypervolume - hv_curve
    if key in generator_curves:
      generator_curves[key] = np.vstack([generator_curves[key], regret_curve])
    else:
      generator_curves[key] = regret_curve

fig, ax = plt.subplots(1, 1, figsize=(4,4))
ax.set_xlabel('# of Pareto Points')
ax.set_ylabel('Hypervolume Regret')
ax.set_title('z =  (3- exp(x)) * (cos(pi * y) + 1)')
ax.set_yscale('log')
for key, curves in generator_curves.items():
  print(curves.shape)
  plot_median_convergence(ax, curves, label=key, percentiles=((25, 75),))
plt.legend()
# -*- coding: utf-8 -*-
"""Multiobjective Linear Bandits

Automatically generated by Colab.

Original file is located at
    https://colab.corp.google.com/drive/1CD7ek1DV4f3FNzoO7H1rmb0kOkMvx80Z
"""

import dataclasses
import itertools
import math
import os
import re

from absl import flags
import matplotlib.pyplot as plt
import numpy as np
# @title Simple LinUCB implementation { display-mode: "form" }
import numpy as np
import pandas as pd
import seaborn as sns

from google3.file.recordio.python import recordio
from google3.learning.vizier import benchmark_v2
from google3.learning.vizier.benchmark_v2 import analysis
from google3.learning.vizier.benchmark_v2 import config_pb2
import google3.pyglib.gfile as gfile

class LinUCB:
  """Simple LinUCB implementation."""

  def __init__(
      self,
      dimension: int,
      max_inst_regret: float = 2.0,
```

```python
        regularizer: float = 1.0,
        var_noise: float = 1.0,
        max_parameter_norm: float = 1.0,
        failure_probabilty: float = 0.1,
    ):
        assert regularizer >= 0.0

        self.dimension = dimension
        self._regularizer = regularizer
        self._var_noise = var_noise
        self._max_parameter_norm = max_parameter_norm
        self._max_inst_regret = max_inst_regret
        self._failure_probability = failure_probabilty

        self.reset()

    def reset(self):
        self._covariance_inv = np.eye(self.dimension) * self._regularizer
        self._reward_scaled_features = np.zeros(self.dimension)
        self._parameter_estimate = np.zeros(self.dimension)
        self._num_observations = 0
        self._last_conf_radius = None
        self._conf_ellipsoid_width_raw = self._conf_ellipsoid_rhs()

    def add_observation(self, action, reward: float, context=None):
        """add an observation (action, reward) to the learner.

        This function updates the covariance matrix and parameter estimate.

        Args:
            action: action that was taken
            reward: achieved reward
            context: context for this observation
            blamed: whether this algorithm is to be blamed for this observation.
        """

        self._num_observations += 1

        # Sherman Morrison update of covariance matrix
        y = np.dot(self._covariance_inv, action)
        self._covariance_inv -= np.outer(y, y) / (1 + np.inner(action, y))
        # regression target
        self._reward_scaled_features += reward * action
        # parameter estimate
        self._parameter_estimate = np.dot(
            self._covariance_inv, self._reward_scaled_features
        )
        self._conf_ellipsoid_width_raw = self._conf_ellipsoid_rhs()

    def ucb(self, actions, confidence_scale=1.0):
        """computes the upper-confidence bound for a batch of actions.

        Args:
            actions: A x d matrix
            confidence_scale: scaling factor in front of the bonus prescribed by
                theory

        Returns:
            upper-confidence bound for each A action, A vector
```

```
      """
      rewards = np.dot(actions, self._parameter_estimate)
      var = np.sum(actions * np.dot(actions, self._covariance_inv), axis=1)
      beta = self._conf_ellipsoid_width_raw * confidence_scale
      return rewards, beta * np.sqrt(var)

  def choose_action(self, actions, confidence_scale=1.0):
      """chooses the action that maximizes the upper confidence bound.

      Args:
        actions: A x d matrix, possible actions to choose from
        confidence_scale: scaling factor in front of the bonus prescribed by
          theory

      Returns:
        chosen action (d vector)
      """

      rewards, conf_b = self.ucb(actions, confidence_scale)
      action_index = randargmax(rewards + conf_b)
      return actions[action_index, :]

  def _conf_ellipsoid_rhs(self):
      # from Abbassi-Yadkori et al. 2011
      beta_t = np.sqrt(self._regularizer) * self._max_parameter_norm
      log_dets = -np.log(self._failure_probability)
      log_dets -= self.dimension / 2 * np.log(self._regularizer)
      log_dets -= np.linalg.slogdet(self._covariance_inv)[1] / 2
      beta_t += np.sqrt(2 * self._var_noise * log_dets)

      return beta_t

def randargmax(b):
  """takes the argmax but randomly picks from the set of maximizers."""
  return np.random.choice(np.flatnonzero(b == b.max()))

def linear_scalarizer(acquisitions, weights):
  sum = 0.0
  for acquisition, weight in zip(acquisitions, weights):
    sum += weight * acquisition
  return sum

def chebyshev_scalarizer(acquisitions, weights):
  min = np.inf
  for acquisition, weight in zip(acquisitions, weights):
    min = np.minimum((acquisition - reference) * weight, min )
  return min

def hypervolume_scalarizer(acquisitions, weights):
  min = np.inf
  for acquisition, weight in zip(acquisitions, weights):
    min = np.minimum((acquisition - reference)/ weight, min )
  return min

def uniform_weights():
  weights = np.random.normal(size=num_metrics)
  return abs(weights)/np.linalg.norm(weights)
```

```python
def boxed_linear_weights():
  # Use bounding box.
  u = np.random.uniform(low=1.0, high=3.0, size=num_metrics)
  return u / np.linalg.norm(u, ord=1, keepdims=True)

def boxed_chebyshev_weights():
  lmda = boxed_linear_weights()
  lmda_prime = 1.0/lmda
  return lmda_prime/np.linalg.norm(lmda_prime, ord=1, keepdims=True)

dim = 5
num_metrics = 16
thetas = np.random.normal(size=(num_metrics, dim))

# Anti-correlate thetas.
thetas[0, :] = -thetas[1, :] + np.random.normal(
    size=thetas[1, :].shape, scale=0.01
)
for i in range(int(num_metrics / 2)):
  index = int(2 * i)
  thetas[index, :] = -thetas[index + 1, :] + np.random.normal(
      size=thetas[index + 1, :].shape, scale=0.01
  )
# Renormalize to make sure |theta| = 1
thetas = thetas / np.linalg.norm(thetas, axis=1, keepdims=True)
reference = -2

def run_linear_bandit(thetas, scalarizer, weight_generator, num_rounds=100):
  algs = [LinUCB(dimension=dim) for theta in thetas]

  expected_rewards = np.empty(shape=(num_rounds, num_metrics))
  actions = np.random.normal(size=(1000, dim))
  actions = actions / np.linalg.norm(actions, axis=1)[..., np.newaxis]
  for t in range(num_rounds):

    index = np.random.choice(len(actions))
    action = actions[index]
    expected_reward = np.inner(thetas, action)
    rewards = expected_reward + np.random.normal(size=expected_reward.shape)
    for alg, reward in zip(algs, rewards):
      alg.add_observation(action, reward)

    weights = weight_generator()
    acquisitions = scalarizer([sum(alg.ucb(actions)) for alg in algs], weights)
    assert len(acquisitions) == len(actions)
    index = randargmax(acquisitions)

    action = actions[index]
    expected_reward = np.inner(thetas, action)
    rewards = expected_reward + np.random.normal(size=expected_reward.shape)
    for alg, reward in zip(algs, rewards):
      alg.add_observation(action, reward)

    expected_rewards[t] = expected_reward
  return actions, expected_rewards

scalarizations = {
    'linear-uniform': (linear_scalarizer, uniform_weights),
    'linear-boxed': (linear_scalarizer, boxed_linear_weights),
```

```
        'chebyshev-uniform': (chebyshev_scalarizer, uniform_weights),
        'chebyshev-boxed': (chebyshev_scalarizer, boxed_chebyshev_weights),
        'hypervolume-uniform': (hypervolume_scalarizer, uniform_weights),
        'hypervolume-boxed': (hypervolume_scalarizer, boxed_linear_weights),
}

from vizier.pyvizier import multimetric
from vizier.pyvizier.multimetric import xla_pareto

pareto_algo = xla_pareto.JaxParetoOptimalAlgorithm()

import matplotlib.pyplot as plt

num_rounds = 100
actions, linear_rewards = run_linear_bandit(
    thetas, scalarizer=hypervolume_scalarizer, weight_generator=uniform_weights
)
all_values = np.inner(actions, thetas)
all_frontier = all_values[pareto_algo.is_pareto_optimal(all_values)]
frontier = linear_rewards[pareto_algo.is_pareto_optimal(linear_rewards)]
plt.scatter(all_values[:, 0], all_values[:, 1], color='grey')
plt.scatter(
    all_frontier[:, 0],
    all_frontier[:, 1],
    color='red',
    label='Pareto-optimal points',
    s=30,
)
plt.scatter(
    frontier[:, 0],
    frontier[:, 1],
    color='green',
    label='Discovered Frontier',
    s=30,
)
plt.xlabel('y_1', fontsize=13)
plt.ylabel('y_2', fontsize=13)
plt.title(
    f'Hypervolume scalarizer optimization at T={num_rounds} ', fontsize=18
)
plt.legend()

origin = -2 * np.ones(shape=num_metrics)
num_repeats = 5
fig, ax = plt.subplots(1, 1, figsize=(12, 8))

for key, (scalarizer, weight_generator) in scalarizations.items():
  hvs = []
  for _ in range(num_repeats):
    _, linear_rewards = run_linear_bandit(
        thetas, scalarizer, weight_generator, num_rounds=200
    )
    front = multimetric.ParetoFrontier(
        points=linear_rewards,
        origin=origin,
        cum_hypervolume_base=xla_pareto.jax_cum_hypervolume_origin,
    )
    hvs.append(front.hypervolume(is_cumulative=True))
  y = np.array(hvs)
```

```
  analysis.plot_median_convergence(
      ax, y, xs=np.array(range(y.shape[1])), label=key, percentiles=((30, 70),)
  )
ax.legend()
ax.set_title(f'Cumulative HV from {origin}', fontsize=15)
ax.set_ylabel('hypervolume (HV)')
ax.set_xlabel('# of Trials')
```

