# OpenReview forum: "Optimal Scalarizations for Sublinear Hypervolume Regret"
_NeurIPS.cc/2024/Conference — NeurIPS 2024 poster_

### Official Review · Reviewer_LEgt · 2024-07-10

**Soundness:** 2
**Presentation:** 2
**Contribution:** 2
**Rating:** 5
**Confidence:** 2

**Summary:**

This abstract presents a study on non-linear scalarization techniques for multi-objective optimization, specifically focusing on hypervolume scalarizations with random weights. The authors prove that this approach achieves optimal sublinear hypervolume regret bounds and apply it to multiobjective stochastic linear bandits, deriving improved regret bounds. Their theoretical findings are supported by empirical results demonstrating superior performance of non-linear scalarizations compared to linear alternatives and other standard multiobjective algorithms in various settings.

**Strengths:**

The paper provides both theoretical and empirical analyses of hypervolume scalarization performance.

**Weaknesses:**

First of all, most lemmas are from the reference [1]. The authors should primarily focus on highlighting differences or contributions compared to this reference. Given that the key scalarization method was proposed in [1], simply conducting experiments on synthetic data adds little value, as this was likely already verified in the original proposal.

Besides, the applicability of hypervolume regret to the multi-armed bandit setting is questionable, especially in scenarios with finite action sets. The identification of the optimal arm in such cases needs to be carefully defined.


[1]. Zhang, Richard, and Daniel Golovin. "Random hypervolume scalarizations for provable multi-objective black box optimization." International conference on machine learning. PMLR, 2020.

**Questions:**

See the weakness.

**Limitations:**

The authors need to clarify the contribution first.

---

> ### Author Rebuttal · Authors · 2024-08-05
>
> **Novelty of our paper:** Our paper is focused on describing how fast scalarizations can approximate the Pareto frontier under finite samples even with *perfect knowledge of the Pareto frontier (whitebox setting)* and is first of its kind (see Theorem 7). Specifically, we introduce the notion of the hypervolume regret convergence rate, which is both a function of both the scalarization and the weight distribution and is a guarantee in the worst case for any frontier. This is complementary to the results of Golovin & Zhang, which showed the expected hypervolume regret bounds is $T^{-1/2}$ under the assumption that the scalarizations, with no generalization error, already cover the whole Pareto frontier in expectation. Specifically, note that their bounds in Golovin & Zhang do not even include the $T^{-1/k}$ rate, which we have shown to be tight. In addition to this main contribution, we also include a linear bandits section showcases the improved non-euclidean analysis of the hypervolume regret and the algorithm is primarily introduced as a tool to demonstrate the utility of the hypervolume scalarization in theory and later in experiments. We will make this more clear in contributions section in the introduction.
>
> **Borrowing Previous Work:** We strongly disagree with the assessment that our work is borrowed from the previous work of Golovin and Zhang. Note that only 1 of our results is borrowed from (and attributed to) the previous work (Lemma 5), while all other results are novel, especially Theorem 7 in the whitebox setting. Furthermore, the experiments on synthetic data were not done before, as the notion of hypervolume convergence rate (in the whitebox setting) was not introduced in the previous paper, which focuses on blackbox optimization and derives a regret bound rate of $T^{-1/2}$.
>
> As mentioned in the introduction, we emphasize that our derived regret rate of the Hypervolume scalarization holds regardless of the multi-objective function or the underlying optimization algorithm; furthermore, we believe these agnostic rates can be a general theoretical tool to compare and analyze the effectiveness of proposed scalarizers (see our experiments on synthetic Pareto frontiers). We will make this distinction clearer.
>
> **Multi-armed Bandit:** Lastly, we do not mention the multi-armed bandit setting in our paper and we focus on only continuous action sets.

---

> > ### Comment · Reviewer_LEgt · 2024-08-10
> >
> > Thank you for your responses and clarification. I changed my rating.

---

### Official Review · Reviewer_ooQr · 2024-07-12

**Soundness:** 3
**Presentation:** 3
**Contribution:** 2
**Rating:** 6
**Confidence:** 3

**Summary:**

This paper shows that the hypervolume scalarization has sublinear hypervolume regret bounds of $O(T^{-1/k})$, and further proves a lower bound of hypervolume regret of $\Omega(T^{-1/k})$. An optimization algorithm for multiobjective linear bandits is proposed. An empirical study is conducted to verify the effectiveness of hypervolume scalarizations.

**Strengths:**

* Although this paper is technical, it is well-organized and easy to follow.
* The proposed theoretical results are interesting and non-trivial.

**Weaknesses:**

* This paper has a significant overlap with a previous paper (Golovin and Zhang, 2020), which proposed hypervolume scalarization and its hypervolume regret. The main focus of these two papers seems similar, so I suggest the authors summarize the differences and new contributions.
* The authors claim that the concept of the so-called "hypervolume scalarization" originated from (Golovin and Zhang, 2020). However, similar scalarization methods were already proposed in (Qi et al., 2014) and have been widely used in the field of multi-objective optimization (Li et al., 2016).
* The authors regard hypervolume as the gold standard, but I have reservations about this. Hypervolume has two main drawbacks: (1) the hypervolume optimal distribution may not cover the entire Pareto Front, and (2) HV's behavior is highly dependent on the choice of the reference point. Therefore, in practical applications, maximizing HV does not necessarily yield ideal results. Additionally, the paper assumes reference points multiple times, such as in Theorem 8 where it is assumed that $z=0$. Given the characteristics of HV, I would question whether such assumptions may limit the generalizability of these theorems.

References

Zhang, R., & Golovin, D. (2020). Random hypervolume scalarizations for provable multi-objective black box optimization. In ICML (pp. 11096-11105).

Qi, Y., Ma, X., Liu, F., Jiao, L., Sun, J., & Wu, J. (2014). MOEA/D with adaptive weight adjustment. Evolutionary Computation, 22(2), 231-264.

Li, H., Zhang, Q., & Deng, J. (2016). Biased multiobjective optimization and decomposition algorithm. IEEE Transactions on Cybernetics, 47(1), 52-66.

**Questions:**

P2, L74. "maximize the Pareto front"?

P3, L130. Typo: "biojective".

**Limitations:**

No concerns.

---

> ### Author Rebuttal · Authors · 2024-08-05
>
> We thank the reviewer for the helpful review.
>
> **Novelty of our paper:** Our paper is focused on describing how fast scalarizations can approximate the Pareto frontier under finite samples even with *perfect knowledge of the Pareto frontier (whitebox setting)* and is first of its kind (see Theorem 7). Specifically, we introduce the notion of the hypervolume regret convergence rate, which is both a function of both the scalarization and the weight distribution and is a guarantee in the worst case for any frontier. This is complementary to the results of Golovin & Zhang, which showed the expected hypervolume regret bounds is $T^{-1/2}$ under the assumption that the scalarizations, with no generalization error, already cover the whole Pareto frontier in expectation. Specifically, note that their bounds in Golovin & Zhang do not even include the $T^{-1/k}$ rate, which we have shown to be tight. We will make this more clear in contributions section in the introduction.
>
> As mentioned in the introduction, we emphasize that our derived regret rate of the Hypervolume scalarization holds regardless of the multi-objective function or the underlying optimization algorithm; furthermore, we believe these agnostic rates can be a general theoretical tool to compare and analyze the effectiveness of proposed scalarizers (see our experiments on synthetic Pareto frontiers). In addition to this main contribution, we also include a linear bandits section showcases the improved non-euclidean analysis of the hypervolume regret and the algorithm is primarily introduced as a tool to demonstrate the utility of the hypervolume scalarization in theory and later in experiments. We will make these contributions clearer in the paper.
>
> **Scalarization Origination:** The hypervolume scalarization origination is indeed difficult to pinpoint but we mainly cite Golovin and Zhang as it contains theoretical analysis of the scalarization, specifically its relation to hypervolume, which is novel to our knowledge. We recognize in our paper that “the classical Chebyshev scalarization has been shown to be effective in many settings, such as blackbox optimization”, and acknowledge the relationship between the two: the Chebyshev scalarization with an appropriate “inverse” weight distribution enjoys the same convergence rate as the hypervolume scalarization. We will add more discussion and the recommended citations to the other similar scalarization methods.
>
> **Hypervolume Standard:** The hypervolume as a standard metric is classically used because it does cover the full Pareto frontier, as the hypervolume metric has strict Pareto compliance meaning that if a set $A \subseteq B \subset R^k$, then the hypervolume of $B$ is greater than that of $A$ [see Golovin & Zhang]. We will clarify this in our final version.
>
> **Reference Point:** We acknowledge that the choice of the reference point can be a bit tricky in practice but many packages use reasonable heuristics and we find that our experiments are not sensitive to this choice [see Daulton et. al. 2020].  In theory, our assumptions on reference point are held without loss of generality and our hypervolume convergence bounds hold for ANY choice of reference point. Therefore, for Theorem 8, we only need to find a specific counterexample to establish a lower bound and so we choose $z = 0$, but the generalization is in fact not affected, as the lower bound can be extended to any reference point by shifting the outputs. We will clarify the role of the reference point more in the final version.
>
> We fixed the typos, thank you, and hope that our clarifications have improved the reviewer's confidence about the quality of our work.
>
> Refs:
>
> Daulton et. al 2020. Differentiable Expected Hypervolume Improvement for Parallel Multi-Objective Bayesian Optimization

---

> > ### Comment · Reviewer_ooQr · 2024-08-09
> >
> > Thank you for your response. I am satisfied with the response, and I have increased my rating. Regarding your response to HV, I think HV covers the full PF only under certain assumptions. First, the reference point is set appropriately. Second, there are infinite solutions.

---

> > > ### Author Response · Authors · 2024-08-09
> > >
> > > Thank you for the update! We agree that the reference point needs to be set sanely (heuristics is usually set reference to min - 0.1 * range), but for the second point, not sure if the number of solutions matter. Happy to entertain a longer discussion if there's still concern on the latter.

---

### Official Review · Reviewer_3k2M · 2024-07-15

**Soundness:** 2
**Presentation:** 3
**Contribution:** 3
**Rating:** 6
**Confidence:** 2

**Summary:**

For multi-objective optimization, a common technique is to use scalarizations to reduce the multi-objective to one single objective. While it is easy to use linear function/scalarizations, it does not fully explore a concave region of Pareto frontier. The paper focuses on non-linear scalarization, in particular, it focuses on hypervolume scalarizations.

The authors propose simple non-linear scalarizations that effectively explore the Pareto frontier and achieve optimal sublinear hypervolume regret bound.

The main contribution of this paper is to show that hypervolume scalarizations with uniformly random weights achieves a sublinear hypervolume regret bound of order $O(T^{-1/k})$ together with a matching lower bound.

For the special case of multiobjective linear bandits, the paper gives an algorithm that achieves $O(dT^{−1/2} + T^{1/k})$ hypervolume regret.

The paper also empirically justifies the effectiveness of hypervolume scalarizations via synthetic, linear bandit, and blackbox optimization benchmarks.



--------------
I acknowledge that I have read the author's rebuttal.

**Strengths:**

The multi-objective optimization task is very common and the technique scalarization seems to be very important (both practical and theoretically). The paper explores non-linear scalarization technique and gives sublinear regret bound.

**Weaknesses:**

.

**Questions:**

.

**Limitations:**

One minor suggestion to improve the readability of this paper is to give some concrete examples on scalarization, and in particular, for hyper volume regret.

---

> ### Author Rebuttal · Authors · 2024-08-05
>
> We thank the reviewer for the helpful review.
>
> **Scalarizations and Hypervolume Regret:** Our study considers both the scalarization and the weight distribution (see Figure 1 for visualization) and provides a regret guarantees in the worst case for all frontiers. We have included some novel synthetic experiments that gives a complete whitebox characterization of the Pareto frontiers and explicitly calculates the hypervolume regret given our optimization procedure using different scalarizations. We hope to make this more clear in our final draft with better visualization.

---

### Official Review · Reviewer_9Trh · 2024-07-18

**Soundness:** 3
**Presentation:** 2
**Contribution:** 3
**Rating:** 7
**Confidence:** 2

**Summary:**

The paper addresses the challenge of exploring the Pareto frontier in multi-objective optimization problems, particularly focusing on minimizing hypervolume regret. Linear scalarizations are often inadequate as they fail to explore certain non-convex regions of the Pareto frontier. The authors propose using non-linear scalarization with weights chosen uniformly at random. Specifically, the scalarization function $s_\lambda(y) =  \min_{i \in [k]} (y_i/\lambda_i)^k$, where $y$ is a $k$-dimensional vector and $\lambda$ is chosen uniformly at random. This achieves optimal sublinear hypervolume regret bounds of $O(T^{-1/k})$ with matching lower bounds.  Furthermore, the authors show that hypervolume scalarization uniformly explores the Pareto frontier in terms of the angular direction, thereby covering the entire Pareto frontier. They further demonstrate the theoretical and empirical performance of these scalarizations across various optimization settings. For multi-objective linear bandits, the authors introduce a scalarized algorithm based on the UCB algorithm and prove that this algorithm achieves a regret bound of $O(d \sqrt{T} + T^{-1/k})$.

**Strengths:**

The paper makes relevant contributions to multi-objective optimization literature. The setting itself is well justified, and the authors thoroughly compare their contributions to previous work. Although the scalarization functions are not novel, given the prior work, the results presented in this work make notable advances.

In particular, Lemma 5 offers good insight into why hypervolume scalarization is potentially a good strategy even for complicated optimization problems. The fact that hypervolume scalarization uniformly explores the Pareto frontier (in an angular sense) ensures that hypervolume regret is small. Moreover, while it is known that the expected value of the scalarization function (with uniform exploration) gives the hypervolume, this result is asymptotic, and Theorem 7 quantifies the actual convergence rate (with finite samples). Additionally, the lower bound in Theorem 8 establishes that the convergence rate is optimal.

**Weaknesses:**

Please see the questions section.

**Questions:**

The paper briefly discusses Chebyshev scalarization (which has a similar form except that $\lambda_i$ is in the numerator), mentioning that it enjoys similar advantages as hypervolume scalarization when dealing with a non-convex Pareto frontier. However, it is not clear how the hypervolume regret would scale for this function. The paper could benefit from a theoretical comparison with other scalarizations as well.

In the case of linear bandits, the UCB extension works for other scalarization functions (Chebyshev, linear) as seen in the algorithm and achieves sublinear regret as shown in the experiments. I wonder if the theoretical guarantees differ significantly for different scalarization functions. What exact properties of the scalarization function impact the regret?

**Limitations:**

This paper does not have any negative social impact.

---

> ### Author Rebuttal · Authors · 2024-08-05
>
> We thank the reviewer for the helpful review.
>
> **Comparison with Other Scalarizations:** The hypervolume regret convergence rate is both a function of both the scalarization and the weight distribution and is a guarantee in the worst case for all frontiers in the whitebox setting. Even the hypervolume scalarization, with a skewed weight distribution, stands no chance of a sublinear hypervolume convergence rate. Therefore, we need to compare theoretical convergence rates using the scalarization along with the weight distribution.
>
> As you stated, we have mentioned that the Chebyshev scalarization with an appropriate “inverse” weight distribution enjoys the same convergence rate as the hypervolume scalarization. But to your point, we hope to add a theoretical analysis of the worst case convergence rate of the Chebyshev scalarization with a uniform weight distribution, which can likely be done via a KL analysis of the weight distribution.
>
> As for the linear scalarization, since hypervolume convergence is a worst case measure, note that any scalarization that uses a linear component will not be able to achieve sublinear hypervolume regret rates (regardless of weight distribution), as it will not be able to explore the whole Pareto frontier in adversarially designed concave settings.
>
> **Scalarization on Regret:** The bandits section showcases the improved non-euclidean analysis of the hypervolume regret and the algorithm is primarily introduced as a tool to demonstrate the utility of the hypervolume scalarization in theory and later in experiments. The theoretical guarantees for linear bandit regret bounds are based on the $\ell_p$ smoothness of the scalarization function, where $p$ is the norm chosen for the artifact of analysis.
>
> The bounds can differ significantly as the number of objectives $k$ increases, as the Hypervolume scalarization has an $\ell_\infty$ smoothness that is independent of $k$ whereas this is not true of other scalarizations, which grow polynomially in $k$. In our experiments, we do find that as $k$ increases, we have more advantage in using the Hypervolume scalarization, over other scalarizations. We will make this more clear in the final draft.

---

> > ### Comment · Reviewer_9Trh · 2024-08-13
> >
> > Thank you for your response. I have no further questions.

---

### Decision · Program_Chairs · 2024-09-25

**Decision:**

Accept (poster)

**Comment:**

This paper proposes a non-linear scalarization with weights chosen uniformly at random to reduce multiple objectives into one. A sublinear hypervolume regret bound of O(T^{-1/k}) is achieved, as well as a matching lower bound. Scalarization in multi-objective optimization is an important problem. The reviewers found the results and techniques interesting and non-trivial.